# Exploring Polyglot Harmony: On Multilingual Data Allocation for Large Language Models Pretraining

**Ping Guo**[‡]    **Yubing Ren**[†]    **Binbin Liu**[‡]    **Fengze Liu**[‡]
**Haobin Lin**[‡]  **Yifan Zhang**[‡]  **Bingni Zhang**[‡]  **Taifeng Wang**[‡]  **Yin Zheng**[‡*]
[‡]ByteDance       [†]Institute of Information Engineering, Chinese Academy of Sciences

## Abstract

Large language models (LLMs) have become integral to a wide range of applications worldwide, driving an unprecedented global demand for effective multilingual capabilities. Central to achieving robust multilingual performance is the strategic allocation of language proportions within training corpora. However, determining optimal language ratios is highly challenging due to intricate cross-lingual interactions and sensitivity to dataset scale. This paper introduces CLIMB (**C**ross-**L**ingual **I**nteraction-aware **M**ultilingual **B**alancing), a novel framework designed to systematically optimize multilingual data allocation. At its core, CLIMB introduces a *cross-lingual interaction-aware language ratio*, explicitly quantifying each language's effective allocation by capturing inter-language dependencies. Leveraging this ratio, CLIMB proposes a principled two-step optimization procedure—first equalizing marginal benefits across languages, then maximizing the magnitude of the resulting language allocation vectors—significantly simplifying the inherently complex multilingual optimization problem. Extensive experiments confirm that CLIMB can accurately measure cross-lingual interactions across various multilingual settings. LLMs trained with CLIMB-derived proportions consistently achieve advanced multilingual performance, even achieve competitive performance with open-sourced LLMs trained with more tokens.

## 1   Introduction

Large language models (LLMs), exemplified by the GPT series [42, 43], LLaMA series [61, 60, 25], Gemma series [20, 19, 21], Qwen series [48, 49, 58], and DeepSeek series [14, 13], have reshaped various language-based applications worldwide, powering advanced chatbots [12], machine translation systems [70], and intelligent virtual assistants [62]. Such impressive capabilities emerge predominantly from extensive pretraining on enormous textual datasets, frequently spanning tens to hundreds of trillions of tokens, enabling the capture of rich and diverse linguistic knowledge. Driven by the growing global demand and the need for equitable language representation, there has been an accelerating shift toward multilingual pretraining, aiming to transcend linguistic boundaries and serve a broader range of linguistic communities effectively [69]. Central to this shift lies a fundamental question: **how should the proportions of different languages be optimally allocated within the training corpus to achieve balanced and superior model performance across all target languages?**

However, determining an optimal multilingual mixture poses considerable challenges. The foremost difficulty arises from cross-lingual interactions: performance on one language can be significantly influenced by other languages trained concurrently [16, 7]. As illustrated in Figure 1, even when the training proportion of Arabic remains fixed to 10%, modifying the proportions of the other four

---

*Yin Zheng (yzheng3xg@gmail.com) is the corresponding author and tech lead of the multilingual LLM pretrain project.

languages (increasing one language to 60% proportion) in a five-language LLM can substantially alter Arabic's performance. This interdependence prevents isolated optimization of individual languages and necessitates joint optimization of the entire language set. Additionally, optimal language ratios are sensitive to the scale of the training corpus [32, 28, 55, 24]. Specifically, language proportions identified as optimal at smaller scales (e.g., 1 billion tokens) may no longer remain optimal when scaled to larger training sets (e.g., 4 trillion tokens), rendering simple extrapolations unreliable and incurring prohibitive experimental costs. Consequently, current multilingual LLMs often resort to heuristic trial-and-error approaches [15, 25], or reuse language ratios derived from prior models without systematic justification [34], highlighting a critical need for a principled and scalable solution to multilingual data allocation.

In pursuit of achieving the optimal language allocation, this paper explores whether it is possible to accurately predict model performance under various language allocations without explicitly training the models. Inspired by the concept of scaling laws, which characterize how a model's validation loss systematically scales with model size ($N$) and data volume ($D$) [33, 30], we hypothesize that a similar predictive framework could be applied to multilingual settings by incorporating language proportions. Specifically, if we can formulate a mathematical relationship that captures how validation performance varies with language proportions in the training corpus, then it becomes feasible to infer optimal language ratios by identifying the allocations that minimize validation loss. However, due to the intricate cross-lingual interactions

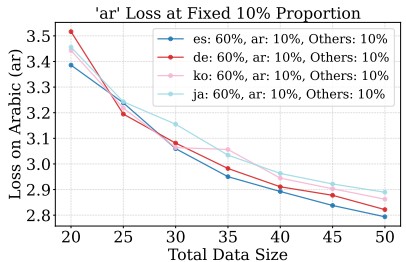

Figure 1: Cross-lingual Interactions in a five-langauge LLM.

among languages, precisely modeling and predicting validation performance across different language compositions remains highly challenging.

In this paper, we propose **CLIMB (Cross-Lingual Interaction-aware Multilingual Balancing)**, a novel framework designed to systematically optimize language proportions for multilingual LLM pre-training. Our approach consists of two interconnected components. First, we introduce the *cross-lingual interaction-aware language ratio*, a novel metric that explicitly quantifies the effective allocation of each language in the presence of cross-lingual interactions, effectively reflecting the impact of other jointly trained languages. Second, leveraging these cross-lingual interaction-aware ratios, we can estimate the optimal multilingual balance by decomposing the optimization into two steps: initially, we determine the direction of optimal allocation by equalizing the marginal benefits across languages; subsequently, we obtain the estimated optimal proportions by maximizing the magnitude of the resulting cross-lingual interaction-aware language ratio vector. This principled two-step procedure enables efficient and accurate computation of multilingual data distributions, significantly reducing the complexity inherent in direct joint optimization.

To comprehensively evaluate the effectiveness of CLIMB, we conduct experiments in two primary aspects. First, we validate the predictive accuracy of the proposed cross-lingual interaction-aware language ratio. By integrating this novel ratio into the multilingual scaling law framework, we observe a substantial improvement in predictive accuracy compared to baseline scaling laws relying on independence assumptions among languages. Second, leveraging the optimal proportions computed via CLIMB, we train multilingual LLMs at both 1.2B and 7B parameter scales. Experimental results demonstrate that models pretrained with CLIMB-derived ratios consistently achieve leading performance compared to various baselines with alternative language allocations. Remarkably, even compared to open-sourced models pretrained on more tokens, our CLIMB-optimized models exhibit highly competitive performance across multiple multilingual benchmarks.

## 2 CLIMB

Our approach is grounded in extensive multilingual experiments designed to disentangle how loss dynamics evolve with respect to language composition, total training tokens, and cross-lingual proportions. Building on these observations, our framework consists of two main components: the *Cross-lingual Interaction-aware Language Ratio*, which explicitly models effective language proportions by incorporating inter-language dependencies, and the *Optimal Multilingual Balance*,

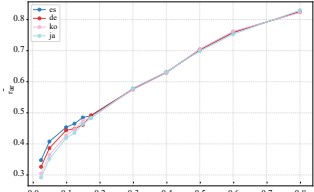 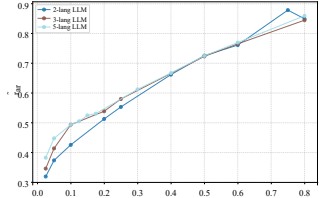 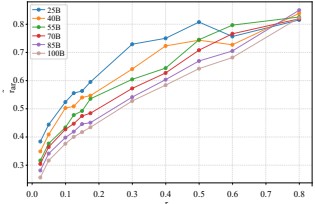

(a) $\tilde{r}_{ar}$ as a function of $r_{ar}$ with varying co-training languages

(b) $\tilde{r}_{ar}$ as a function of $r_{ar}$ with varying language counts

(c) $\tilde{r}_{ar}$ as a function of $r_{ar}$ with varying training data scales

Figure 2: Illustration of cross-lingual interaction-aware language ratio ($\tilde{r}_{ar}$) and its dependency on original training proportions ($r_{ar}$).

which leverages these interaction-aware ratios to estimate the optimal allocation $\mathbf{r}^*$ that minimizes multilingual validation loss.

## 2.1 Experimental Setup

To study how multilingual training dynamics depend on language composition, token scale, and inter-language proportions, we conduct a series of controlled experiments across diverse language settings, organized along three dimensions:

**(1) Number of Languages.**

We explore multilingual configurations of increasing scope, including **bilingual** ({es–ko, en–zh, de–ar, ko–ja}), **trilingual** ({es–de–ar, es–ko–zh, en–zh–ja}), **five-language** ({es–de–ar–ko–ja}), and a **sixteen-language** setting covering {de, en, nl, es, pt, fr, it, id, ja, ko, zh, ru, ar, th, vi, tr}.

**(2) Total Training Tokens.** For each setting, models are trained under ten token budgets from 5B to 50B (step size 5B). To ensure comparability, all runs share the same learning rate schedule, decaying to 10% of the initial rate by training end.

**(3) Language Proportion.** To examine proportional effects, one language's share is fixed while others evenly split the remainder. For each language $L_i$, its proportion is varied over {0.02, 0.025, 0.05, 0.1, 0.2, 0.25, 0.4, 0.5, 0.6, 0.75, 0.8, 0.9, 0.95, 0.975, 0.98} to observe loss trends.

Combining these factors yields over 500 multilingual runs. Section 2 summarizes empirical findings and fitting equations, while Section 3 examines their extrapolation and generalization performance.

## 2.2 Problem Formulation

Given a multilingual corpus consisting of training data from $m$ distinct languages $L_1, \ldots, L_m$, our objective is to determine the optimal language allocation for pretraining LLMs. Formally, we define the language proportion vector as $\mathbf{r} = [r_1, r_2, \ldots, r_m]^\top \in \mathcal{R}^m$, where $\mathcal{R}^m = \{\mathbf{r} \in \mathbb{R}^m \mid \sum_{i=1}^m r_i = 1, r_i \geq 0, \forall i\}$ denotes the probability simplex.

Given a total token budget $D$, each language $L_i$ contributes $D_i = \lfloor r_i \cdot D \rfloor$ tokens to the training set. The model parameters $\theta$ are trained via empirical risk minimization: $\theta^*(D, \mathbf{r}) = \arg\min_\theta \mathcal{L}(\theta; D, \mathbf{r})$, where $\mathcal{L}(\theta; D, \mathbf{r})$ denotes the next-token prediction loss on the multilingual training set defined by proportions $\mathbf{r}$ and token budget $D$.

To evaluate the pretrained model, we measure validation loss on a language-specific held-out set $D_i^v$: $\mathcal{L}_i^v(\theta^*(D, \mathbf{r})) = \mathcal{L}(\theta^*(D, \mathbf{r}); D_i^v)$.

Our goal is to identify the optimal language proportion vector $\mathbf{r}^*$ that achieves balanced multilingual performance by minimizing a weighted sum of validation losses across all languages:

$$\mathbf{r}^* = \arg\min_{\mathbf{r} \in \mathcal{R}^m} \sum_{i=1}^m \omega_i \cdot \mathcal{L}_i^v(\theta^*(D, \mathbf{r})), \tag{1}$$

where hyperparameter $\omega_i \geq 0$ specify the relative importance of each language $L_i$, set according to application-specific requirements or practical considerations.

This formulation defines a bi-level optimization problem, in which the outer optimization seeks optimal language proportions, and the inner optimization involves training an LLM given these language proportions. Due to the intrinsic complexity of cross-lingual interactions and the prohibitive computational cost of repeated model retraining, directly solving this optimization problem through standard gradient-based approaches is computationally infeasible.

### 2.3 Cross-Lingual Interaction-aware Language Ratio

Given a total token budget $D$ and a language proportion vector $\mathbf{r}$, we can obtain the validation loss $\mathcal{L}_i^v(D, \mathbf{r})$ for language $L_i$. Let $\tilde{D}_i$ be the number of tokens required to reach the validation loss $\mathcal{L}_i^v(D, \mathbf{r})$ from a monolingual model solely on language $L_i$, we then define the **cross-lingual interaction-aware ratio** $\tilde{r}_i$ as the ratio of this equivalent monolingual token budget $\tilde{D}_i$ to the actual multilingual token budget $D$: $\tilde{r}_i = \frac{\tilde{D}_i}{D}$. Formally, $\tilde{r}_i$ can be formally expressed as:

$$\tilde{r}_i = \frac{1}{D} \left( \frac{B_i}{\mathcal{L}_i^v(D, \mathbf{r}) - E_i} \right)^{1/\beta_i}, \tag{2}$$

where parameters $B_i$, $\beta_i$, and $E_i$ are derived from the monolingual scaling law [30]. Specifically, the monolingual scaling law characterizes how the validation loss decreases as training data volume increases for a single language $L_i$, expressed as:

$$\mathcal{L}_i^v(D_i, r_i = 1) = \frac{B_i}{D_i^{\beta_i}} + E_i, \tag{3}$$

where $D_i$ represents the token budget allocated exclusively to language $L_i$. In the absence of cross-lingual transfer, the interaction-aware ratio $\tilde{r}_i$ equals the actual ratio $r_i$, thus the difference $\tilde{r}_i - r_i$ quantifies the magnitude of cross-lingual effects from other languages.

#### 2.3.1 Empirical Observations and Insights

To systematically understand the behavior of the cross-lingual interaction-aware language ratio $\tilde{r}_i$, we conducted over 300 experiments on Transformer-based models. Specifically, we varied the number of jointly trained languages (2, 3, and 5 languages), total token budgets ranging from 5 billion to 100 billion tokens, and explored a wide range of language proportion vectors $\mathbf{r}$. For each configuration, we computed the pairs $(r_i, \tilde{r}_i)$ to examine how the effective language ratio deviates from the actual proportion due to cross-lingual transfer. These results are visualized in Figure 2, from which we identify following key empirical insights:

- **Dependency on absolute language proportion.** Cross-lingual transfer strength diminishes as the actual language proportion ($r_i$) increases, with the slope gradually decreasing and approaching linearity at higher proportions, as illustrated in Figures 2 (a), (b), and (c).

- **Dependency on co-training languages.** The specific set of co-training languages affects cross-lingual transfer primarily when the language proportion $r_i$ is small, as demonstrated in Figure 2 (a). This influence diminishes as $r_i$ grows.

- **Dependency on model language counts.** Increasing the number of co-trained languages affects the intercept rather than the slope of the cross-lingual transfer relationship. This variation shifts the onset point at which transfer strength approaches linearity, as shown in Figure 2 (b).

- **Dependency on data scale.** Cross-lingual transfer consistently weakens with larger total token budgets ($D$), indicating that increased training data volume reduces dependency between languages, as depicted in Figure 2 (c).

These patterns are consistent with prior findings on the *curse of multilinguality* [3, 10], which similarly report reduced transfer when auxiliary-language data dominates and when model capacity is spread across too many languages.

### 2.3.2 Parametric Modeling of Cross-Lingual Interaction-aware Ratio

Motivated by the empirical insights described above, we propose a parametric model to capture the relationship between the cross-lingual interaction-aware language ratio $\tilde{r}_i$ and the actual language ratio $r_i$. Specifically, we model $\tilde{r}_i$ as:

$$\tilde{r}_i = r_i + \left( \sum_{j \neq i} \alpha_{j \to i}(D) \cdot r_j \right) \left( 1 - e^{-\eta_i r_i} \right), \tag{4}$$

where the parameters are defined as follows:

- $\alpha_{j \to i}(D)$ represents the transfer strength from language $L_j$ to language $L_i$. Empirically, we find that this transfer effect diminishes linearly with increasing token budget $D$, which we model as: $\alpha_{j \to i}(D) = b_{ji} + \frac{k_{ji}}{D}$, where $b_{ji}$ indicates the initial strength of cross-lingual transfer from $L_j$, and $k_{ji}$ quantifies the rate at which this transfer strength decays as data volume increases. Details about $\alpha_{j \to i}(D)$ is in Appendix A.

- $\eta_i$ captures the intrinsic data sufficiency of language $L_i$. A larger value of $\eta_i$ indicates that language $L_i$ remains reliant on cross-lingual transfer across a wider range of proportions, exhibiting a pronounced curved (transfer-dominated) regime. Conversely, a smaller $\eta_i$ signals that language $L_i$ quickly enters a linear (self-dominated) regime, reflecting sufficient self-contained data.

### 2.3.3 Complete Cross-Lingual Interaction-aware Scaling Law

By incorporating the parametric definition of the cross-lingual interaction-aware ratio into the monolingual scaling law, we obtain our final scaling law formulation:

$$\mathcal{L}_i^v(D, \mathbf{r}) = \frac{B_i}{[D \cdot \tilde{r}_i]^{\beta_i}} + E_i \tag{5}$$

$$= \frac{B_i}{\left[ D \cdot \left( r_i + \left( \sum_{j \neq i} (b_{ji} + \frac{k_{ji}}{D}) \cdot r_j \right) \cdot (1 - e^{-\eta_i r_i}) \right) \right]^{\beta_i}} + E_i. \tag{6}$$

The complete set of parameters to estimate are: $\{B_i, \beta_i, E_i\}_{i=1}^m$, $\{b_{ji}, k_{ji}\}_{i,j=1, j \neq i}^m$, $\{\eta_i\}_{i=1}^m$.

**Parameter Estimation Procedure.** To fully determine these parameters, we perform targeted experiments involving each language individually. Specifically, for each language $L_i$, we conduct three experiments with distinct proportions: one monolingual scenario (where $r_i = 1$) to estimate the baseline scaling law parameters $B_i$, $\beta_i$, and $E_i$, and two additional multilingual experiments with randomly chosen language proportions $r_i$ and the remaining languages allocated equally as $\frac{1-r_i}{m-1}$. Each of these experiments is repeated at two distinct training token budgets to ensure reliable parameter fitting across data scales. Following the experimental setup [30], we fit our scaling law parameters only using data points from the last 15% of training. Thus, for a setting with $m$ languages, this structured approach requires a total of $3 \times m \times 2$ experiments, enabling comprehensive and accurate estimation of the proposed scaling law parameters. The detailed fitting procedure is summarized in Algorithm 1.

### 2.4 Estimating Optimal Multilingual Balanced Allocation

Directly minimizing the multilingual validation loss defined by Equation (6) is challenging, as it forms a non-convex optimization problem in language proportions $\mathbf{r}$. While it may appear intuitive to directly optimize the cross-lingual interaction-aware language ratios $\tilde{r}_i$ under Equation (6), this objective is intractable in practice, as the total sum $\sum_i \tilde{r}_i$ remains unknown. To address this difficulty, we propose a two-stage optimization procedure that decomposes the original complex problem into two simpler, sequential steps. Specifically, we first determine the optimal direction in the cross-lingual interaction-aware language ratios $\tilde{r}_i$ space, ensuring balanced marginal benefits across languages. Subsequently, we optimize the magnitude along this determined direction to identify the final allocation $\mathbf{r}$ that maximizes the overall cross-lingual interaction-aware language ratios $\tilde{r}_i$, effectively minimizing the multilingual validation loss.

**Algorithm 1** CLIMB

**Input:** Languages $\{L_1, \ldots, L_m\}$, token budgets $\{D^{(1)}, D^{(2)}\}$.
**Output:** Parameters $\{B_i, \beta_i, E_i\}_{i=1}^m$, $\{b_{ji}, k_{ji}\}_{i,j=1,j\neq i}^m$, $\{\eta_i\}_{i=1}^m$, optimal language proportions $\mathbf{r}^*$.

*Part I: Parameter Modeling of Cross-Lingual Interaction-aware Language Ratio*

1: **for** each language $L_i$ **do**
2:    Conduct monolingual experiments ($r_i = 1$) at $D^{(1)}, D^{(2)}$.
3:    Fit monolingual scaling law 3 to estimate $B_i, \beta_i, E_i$.
4:    **for** each proportion $r_i = c_i \in (0, 1)$, repeat twice **do**
5:       Set other languages proportion $r_j = \frac{1-c_i}{m-1}, \forall j \neq i$.
6:       **for** each token budget $D \in \{D^{(1)}, D^{(2)}\}$ **do**
7:          Train model with proportions $\mathbf{r}$ and budget $D$.
8:          Record validation loss $\mathcal{L}_i^v(D, \mathbf{r})$.
9:          Compute $\tilde{r}_i$ from Eq. (6).
10:       **end for**
11:    **end for**
12:    Fit parameters $b_{ji}, k_{ji}, \eta_i$ using $(r_i, \tilde{r}_i)$ pairs.
13: **end for**

*Part II: Estimating Optimal Multilingual Balanced Allocation*

14: Compute optimal direction components $p_i$ via Eq. (7).
15: Normalize direction: $\hat{p}_i \leftarrow p_i / \sum_j p_j$ for all $i$.
16: Solve constrained optimization (Eq. (8)).
17: **return** parameters $\{B_i, \beta_i, E_i\}_{i=1}^m$, $\{b_{ji}, k_{ji}\}_{i,j=1,j\neq i}^m$, $\{\eta_i\}_{i=1}^m$, and optimal proportions $\mathbf{r}^*$.

### 2.4.1 Optimal Direction via Marginal-Benefit Balancing.

In the first stage, we identify the optimal direction for the cross-lingual interaction-aware language ratios $\tilde{r}_i$ by balancing the marginal benefits across all languages. Specifically, we derive the optimal proportional relationship between the interaction-aware ratios by equalizing the marginal validation-loss reduction contributed by each language. The resulting optimal direction $p_i$ for each language $L_i$ is formally given by (see detailed derivation in Appendix B):

$$p_i = \frac{(\omega_i B_i \beta_i)^{1/(\beta_i+1)} D^{-\beta_i/(\beta_i+1)}}{\sum_{k=1}^m (\omega_k B_k \beta_k)^{1/(\beta_k+1)} D^{-\beta_k/(\beta_k+1)}}, \tag{7}$$

where $B_i$ and $\beta_i$ are the monolingual scaling-law parameters of language $L_i$, and $\omega_i$ represents the predefined importance weight for language $L_i$. Intuitively, the direction $p_i$ indicates the ideal relative allocation of interaction-aware language ratios, balancing each language's data efficiency, validation-loss reduction rate, and relative importance. Identifying this optimal direction substantially reduces complexity in subsequent optimization steps by constraining the search space for the final language proportions.

### 2.4.2 Optimal Magnitude via Constrained Effective Allocation Maximization.

With the optimal direction $p_i$ identified, the second stage focuses on determining the optimal magnitude along this direction. Nevertheless, due to the monotonicity of the scaling law function 3, we find that a larger aggregate $\sum_i \tilde{r}_i$ consistently implies a lower overall training loss, thereby revealing an implicit preference for maximizing effective data contributions across languages (details in Appendix C). Specifically, we recover the actual language proportions $\mathbf{r}$ by solving a constrained optimization problem that maximizes the total cross-lingual interaction-aware language ratio while staying close to the previously determined direction $p$. Formally, this optimization objective is defined as:

$$\min_{\mathbf{r}} \left[ -\sum_{i=1}^m \tilde{r}_i(\mathbf{r}) + \rho \sum_{i=1}^m (\hat{r}_i(\mathbf{r}) - p_i)^2 \right], \quad \text{s.t.} \quad \sum_{i=1}^m r_i = 1, \quad r_i \geq 0, \tag{8}$$

where $\hat{r}_i = \frac{\tilde{r}_i}{\sum_j \tilde{r}_j}$ is the normalized interaction-aware ratios and direction components, respectively. The first term of the objective function aims at maximizing the overall interaction-aware language ratio, corresponding directly to minimizing multilingual validation loss, while the second term (soft-constraint) penalizes deviations from the optimal direction $p$. The hyperparameter $\rho > 0$ balances

Table 1: Fitting and Extrapolation Performance of Different Methods ($R^2\uparrow$ and Huber Loss$\downarrow$) for Multilingual LLMs at 100B and 1T Tokens.

| | 2-lang LLM | | 3-lang LLM | | 5-lang LLM | | 16-lang LLM | |
|---|---|---|---|---|---|---|---|---|
| | $R^2\uparrow$ | Huber$\downarrow$ $(\times10^{-3})$ | $R^2\uparrow$ | Huber$\downarrow$ $(\times10^{-3})$ | $R^2\uparrow$ | Huber$\downarrow$ $(\times10^{-3})$ | $R^2\uparrow$ | Huber$\downarrow$ $(\times10^{-3})$ |
| **Fitting Results (Total Training Tokens: 100B)** | | | | | | | | |
| Isolated | 0.649 | 7.95 | 0.743 | 5.35 | 0.734 | 5.34 | 0.768 | 5.26 |
| MSL | 0.832 | 5.61 | 0.854 | 2.15 | 0.823 | 1.94 | 0.836 | 2.20 |
| CLIMB | **0.978** | **0.518** | **0.986** | **0.301** | **0.992** | **0.205** | **0.981** | **0.274** |
| **Extrapolation Results (Total Training Tokens: 1T)** | | | | | | | | |
| Isolated | 0.648 | 8.21 | 0.741 | 5.38 | 0.732 | 5.36 | 0.767 | 5.30 |
| MSL | 0.830 | 5.79 | 0.852 | 2.24 | 0.822 | 1.98 | 0.834 | 2.24 |
| CLIMB | **0.964** | **0.525** | **0.947** | **0.310** | **0.948** | **0.208** | **0.936** | **0.278** |

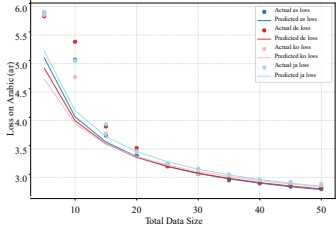

Figure 3: Curve fitting on Arabic data (5-lang LLM). Solid line: fitted results.

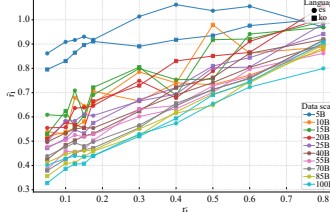

Figure 4: $\tilde{r}$ vs. $r$ across corpus scales on 2 different languages (es, ko).

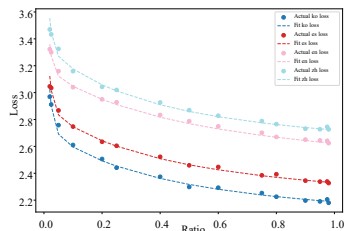

Figure 5: Validation loss fitting across language ratios (0.01–0.99).

these two objectives, with smaller values of $\rho$ emphasizing pure loss minimization and larger values enforcing adherence to the optimal direction.

This problem is inherently non-convex due to the interaction-aware ratio's nonlinearity. Thus, we adopt a Trust-Region Interior-Point Method to efficiently handle this constrained optimization problem. This structured reformulation significantly reduces the complexity and dimensionality of the original allocation problem, accelerating convergence and improving numerical stability.

## 3 Crosslingual Interaction-aware Language Ratio Evaluation

### 3.1 Experimental Setup

**Model Architecture.** We utilize the LLaMA-2 [60] architecture with 1.2 billion parameters, training all models from scratch with randomly initialized weights. All experiments are condcuted on Nvidia H100 GPU cards. To ensure consistency with established scaling-law practices, we follow the Chinchilla configuration and set a fixed number of training steps for each dataset size, adjusting the learning-rate decay schedule to cosine. Validation losses are calculated by averaging the results obtained from the final three training steps.

**Datasets.** All experiments are conducted using data sampled from the Fineweb-2 corpus [45]. To rigorously evaluate our Cross-Lingual Interaction-aware Language Ratio across diverse linguistic scenarios, we conduct experiments involving models trained on 2, 3, 5, and 16 languages, respectively. For each multilingual setting, we vary the token budgets from 5 billion to 100 billion tokens. The specific language compositions and detailed training procedures are documented in the Appendix E.

**Evaluation Metrics & Baseline.** We assess the accuracy of our validation-loss predictions primarily using two metrics: the coefficient of determination ($R^2$) and the Huber loss. The $R^2$ score measures the proportion of variance explained by our fitted scaling-law model, with values closer to 1 indicating

greater predictive accuracy. Additionally, we employ Huber loss, a robust error metric combining properties of mean squared error and mean absolute error, which provides resilience against outliers; lower Huber loss values reflect more accurate predictions.

**Baselines** We compare our approach against two baselines: 1) an assumption of no cross-lingual transfer, where each language's validation loss depends solely on its own proportion, labeled as "isolated"; and 2) a recent multilingual scaling-law study [28], referred to as "MSL".

### 3.2 Scaling Law Fit Accuracy

We present the prediction errors of our proposed scaling law compared to the baseline (MSL) in Table 1, evaluating models trained across various multilingual scenarios (2–16 lang LLMs) with token budgets of 100 B and 1 T tokens. Our cross-lingual interaction-aware approach consistently achieves lower prediction errors compared to the baseline, effectively capturing validation-loss trends in both homogeneous (same language-family) and heterogeneous language settings. From 2-lang LLM to 16-lang LLM, CLIMB's Huber loss remains consistently an order of magnitude lower than both baselines, highlighting the importance and prevalence of cross-lingual transfer effects in multilingual models. Conversely, our scaling-law formulation remains robust and delivers accurate loss predictions even in highly complex multilingual scenarios. We also tried different parametric models to fit and the results are shown in Appendix D

### 3.3 Scaling Law Applicability

To evaluate the robustness of our scaling law, we test its validity across varying training scales and language proportions (see Figure 3 and Figure 4). At smaller scales (below 25 B tokens), prediction accuracy is limited due to unstable cross-lingual transfer. As data increases beyond 25 B tokens, predictions align closely with empirical losses, and extrapolation up to 1 T tokens remains consistent (Table 1). Moreover, bilingual experiments (Figure 5) confirm that our formulation accurately fits validation loss across the full range of language proportions (0.01–0.99), demonstrating its broad applicability to multilingual training.

## 4 Multilingual Balanced Allocation Performance

### 4.1 Experimental Setup

**Model Architecture and Training Setup.** We evaluate multilingual performance by training Transformer models based on the LLaMA-2 [60] architecture at two different scales: 1.2 B and 7 B parameters. All models are trained using the Fineweb-2 corpus, identical to the datasets employed in scaling-law experiments, with each model ingesting a total of 1 T tokens. Peak learning rates are set to $4.3 \times 10^{-5}$ for the 1.2 B model and $3.6 \times 10^{-5}$ for the 7 B model, both following a cosine-decay schedule that decays the learning rate down to 10% of its initial value.

**Baselines.** We compare our proposed CLIMB-derived allocations against two categories of baselines. First, we evaluate against publicly available multilingual models, specifically, LLaMA-3.2-1B [25], GEMMA-3-1B-pt [21], Qwen-3-1.7B-base [58], and XGLM-1.7B [37], whose training data distributions are either open-sourced or reported in official documentation. These models serve as strong multilingual references trained on large or well-documented corpora.

Second, under identical model architecture and data volume constraints, we train models using several alternative language allocation strategies: (1) **Uniform**, which distributes tokens equally across all 16 languages; (2) **Isolated**, derived independently from individual monolingual scaling laws; (3) **MSL**, based on the existing multilingual scaling law formula assuming language-family independence; (4) **Natural**, reflecting each language's original data frequency; (5) **Temperature Sampling (Temp)**, which smooths token allocation via temperature-controlled reweighting of language proportions, we use $T = 0.3$ as baseline; and (6) **UniMax** [8], which maximizes the minimum marginal gain across languages to improve balance under limited total tokens.

**Evaluation Benchmarks.** To comprehensively evaluate CLIMB, we translate several English benchmarks into multilingual to further assess model performance; benchmarks translated by us are marked

Table 2: Performance of CLIMB on 1B models and baselines across 18 multilingual benchmarks. Benchmarks translated by us are marked with ‡. Bold numbers denote the best results among data allocation methods. Standard error is in Appendix I

| | Analytical Reasoning | | | | | | Commonsense Reasoning | | |
|---|---|---|---|---|---|---|---|---|---|
| | MGSM | XARC-E‡ | XARC-C‡ | XTQA‡ | INCLUDE | XGPQA‡ | XCOPA | XSC‡ | XHS‡ |
| **Open Source Multilingual LLM** | | | | | | | | | |
| LLaMA-3.2-1B | 3.89 | 47.19 | 29.55 | 37.73 | 28.48 | 24.91 | 57.07 | 58.13 | 41.52 |
| Qwen3-1.7B | 36.95 | 59.53 | 40.18 | 48.88 | 46.87 | 28.55 | 59.58 | 60.47 | 46.59 |
| Gemma-3-1B-pt | 1.78 | 49.29 | 29.69 | 40.07 | 25.63 | 27.32 | 55.58 | 54.84 | 41.59 |
| XGLM-1.7B | 1.89 | 40.17 | 24.60 | 38.42 | 25.96 | 23.31 | 56.73 | 56.99 | 36.18 |
| **Different Data Allocation Methods** | | | | | | | | | |
| Uniform | 2.07 | 59.76 | 35.41 | 39.62 | 25.12 | 26.18 | 59.49 | 58.99 | 48.12 |
| Isolated | 2.11 | 58.53 | 34.78 | 39.71 | 24.82 | 24.58 | 59.42 | 58.74 | 48.11 |
| Natural | 2.05 | 56.32 | 33.54 | 40.63 | 25.20 | 26.37 | 56.54 | 57.38 | 45.68 |
| MSL | 2.10 | 57.60 | 34.17 | 39.49 | 25.00 | 25.36 | 58.06 | 57.94 | 47.02 |
| Temp | 2.11 | 59.31 | 34.93 | 39.38 | 24.82 | 26.50 | 58.69 | 59.54 | 47.56 |
| UniMax | 2.07 | 59.39 | 35.78 | 39.67 | 25.12 | **27.10** | 59.35 | 58.99 | 48.22 |
| CLIMB | **2.40** | **60.45** | **36.56** | **40.94** | **25.92** | 27.03 | **59.98** | **60.54** | **48.75** |

| | Comprehension | | Linguistic Competence | | Knowledge | | | | Translation |
|---|---|---|---|---|---|---|---|---|---|
| | XNLI | Belebele | MultiBLiMP | XWinograd‡ | GMMLU | CMMLU | JMMLU | VMLU | FLORES |
| **Open Source Multilingual LLM** | | | | | | | | | |
| LLaMA-3.2-1B | 41.29 | 30.69 | 77.66 | 76.08 | 28.74 | 30.30 | 29.84 | 29.09 | 44.14 |
| Qwen3-1.7B | 43.25 | 74.81 | 77.72 | 79.12 | 34.24 | 45.16 | 36.09 | 36.26 | 50.25 |
| Gemma-3-1B-pt | 36.33 | 28.13 | 80.88 | 65.79 | 27.12 | 28.59 | 28.48 | 30.05 | 46.55 |
| XGLM-1.7B | 37.35 | 24.21 | 68.10 | 62.60 | 26.08 | 29.04 | 28.37 | 29.41 | 21.80 |
| **Different Data Allocation Methods** | | | | | | | | | |
| Uniform | 40.08 | 23.30 | 62.24 | 73.34 | 29.05 | **34.79** | 32.47 | 31.44 | 47.51 |
| Isolated | 38.93 | 25.27 | 60.75 | 72.31 | 28.64 | 33.78 | 31.85 | 30.91 | 47.58 |
| Natural | 39.05 | 24.11 | 63.18 | 74.98 | 30.23 | 32.10 | 30.94 | 31.12 | 48.54 |
| MSL | 38.54 | 24.55 | 61.94 | 73.09 | 29.00 | 33.24 | 31.49 | 30.25 | 47.75 |
| Temp | 41.03 | 25.27 | 60.75 | 74.76 | 29.04 | 33.03 | 31.98 | 30.27 | 47.46 |
| UniMax | 40.88 | 23.30 | 62.24 | 74.94 | 31.27 | 33.75 | 32.26 | 31.16 | 49.12 |
| CLIMB | **41.65** | **26.17** | **65.54** | **77.48** | **31.78** | 33.67 | **33.21** | **31.76** | **50.43** |

‡ in Table 2. Specifically, we adopt the following tasks: analytical reasoning (*MGSM* [53], *ARC-Easy/ARC-Challenge* (XARC-E/C)‡ [9], *XTQA* (Cross-lingual TruthfulQA)‡ [36], *INCLUDE* [51], *XGPQA*‡ [50]), commonsense reasoning (*XCOPA* [46], *XStoryCloze* (XSC)‡ [37], *XHellaSwag* (XHS)‡ [67]), comprehension (*XNLI* [11], *Belebele* [1]), linguistic competence (*MultiBLiMP* [31], *XWinograd*(XWG)‡ [41]), and knowledge ( *GMMLU* [54], *CMMLU* [35], $JMMLU^2$, $VMLU^3$), and translation (*FLORES* [57]). Details are provided in Appendix F.

## 4.2 Results

Table 2 reports the main results; per-language scores are in Appendix J. With a 1.2 B model trained on 1 T tokens, we remain competitive with LLaMA-3.2, Gemma-3, and Qwen-3. Against alternative allocations, CLIMB is consistently strong—up to +2.60% on XNLI over isolated allocation and +1.85% on average across all tasks—demonstrating the effectiveness of our multilingual allocation.

**Generalization to Larger Models.** Though our optimal language proportions were initially derived using a 1.2B-parameter model, the methodology generalizes effectively to larger scales. We trained a 7B-parameter model using the same total token budget (1T tokens) and evaluated performance (Figure 6). CLIMB-derived allocations consistently outperform baselines by an average of 3.4%. Results for 7B models appear in Appendix G.

**Allocation Differences Across Methods.** To illustrate differences among allocation strategies, Figure 7 compares language proportions of various methods. CLIMB distinctly allocates higher proportions to languages benefiting most from cross-lingual interactions, unlike baselines (*Isolated*, *MSL*, *Nature*), which either allocate evenly or based solely on single-language characteristics. This focused allocation emphasizes the practical advantage of modeling cross-lingual transfer explicitly.

---

[2] https://github.com/nlp-waseda/JMMLU
[3] https://vmlu.ai/

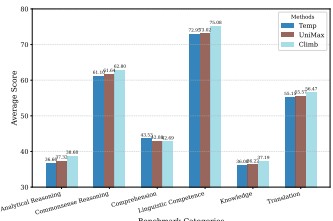
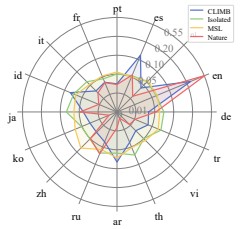
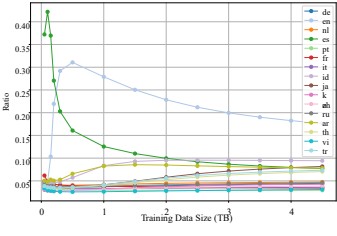

Figure 6: Different allocation results on 7B model.

Figure 7: Illustration of each language allocation methods.

Figure 8: Illustration of how ratios varying with training data.

Table 3: Average cross-lingual transferability per language family. "Transfer-out" measures how much a language benefits others, "Transfer-in" how much it benefits from others. Top1_Lang indicates the strongest transfer source.

| Language | de | en | nl | es | pt | fr | it | id | ja | ko | zh | ru | ar | th | vi | tr |
|---|---|---|---|---|---|---|---|---|---|---|---|---|---|---|---|---|
| Transfer-out | 0.123 | 0.199 | 0.130 | 0.218 | 0.113 | 0.174 | 0.146 | 0.181 | 0.124 | 0.146 | 0.121 | 0.139 | 0.144 | 0.108 | 0.066 | 0.144 |
| Transfer-in | 0.168 | 0.151 | 0.144 | 0.188 | 0.146 | 0.145 | 0.092 | 0.139 | 0.164 | 0.129 | 0.108 | 0.185 | 0.130 | 0.152 | 0.127 | 0.136 |
| Top1_Lang | nl | it | de | pt | es | es | pt | de | zh | ja | ja | tr | it | vi | th | ru |

**Optimal Language Allocation Shifts with Data Scale.** Figure 8 presents how optimal language allocations evolve with increasing token budgets. At smaller scales, simpler or less-resourced languages initially receive higher allocations, quickly lowering validation loss. As data scale increases, allocations shift towards linguistically complex and diverse languages due to their sustained effectiveness at reducing loss. This dynamic trend highlights the necessity of adjusting language proportions according to cross-lingual transfer effects at varying training scales.

**Per-Language and Per-Family Analysis.** We analyze cross-lingual interaction ratios (transfer-out/in) derived from the learned scaling coefficients (Table 3) to explain CLIMB's allocations. Two patterns are clear: (i) strong *intra-family* transfer (e.g., Spanish–Portuguese, Japanese–Korean), and (ii) *high-transfer* languages (e.g., English, Spanish, Indonesian) receive larger shares because their gains generalize broadly. These results improve the interpretability of CLIMB and support its data-driven balancing strategy.

## 5 Related Work

**Data Allocation in Language Model Pretraining.** Recent work optimizes pretraining mixtures at the domain [32, 17], point [63, 66], and token levels [38, 27], typically targeting validation loss. Early approaches use GroupDRO-style reweighting (e.g., DoReMi [64]), while newer methods leverage influence functions and surrogate models [39, 65], gradient approximations [59, 68], and loss-guided heuristics [71, 56] to refine mixtures under budget constraints. However, these techniques are largely monolingual (English), leaving multilingual data allocation comparatively underexplored.

**Scaling Laws for Multilingual LMs.** Scaling laws [33, 29, 30, 40, 47] reliably relate performance to model/data scale and guide efficient allocation. While early work is monolingual, recent studies extend scaling to multilingual settings—mainly in NMT [23, 22, 18, 72, 4, 2, 5, 52, 6]—but typically under simplified bilingual assumptions and encoder–decoder setups. Our formulation makes this interaction explicit for multilingual pretraining.

## 6 Conclusion

This paper introduces CLIMB, a multilingual optimization framework that models cross-lingual interactions to predict optimal language allocations under a scaling-law paradigm. Empirical results show that CLIMB attains strong predictive accuracy and consistently surpasses baselines at both 1.2B and 7B scales. Future work will extend its predictive capacity to languages unseen during training.

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

# A    Details of Fitting Transfer Strength $\alpha_{j \to i}(D)$

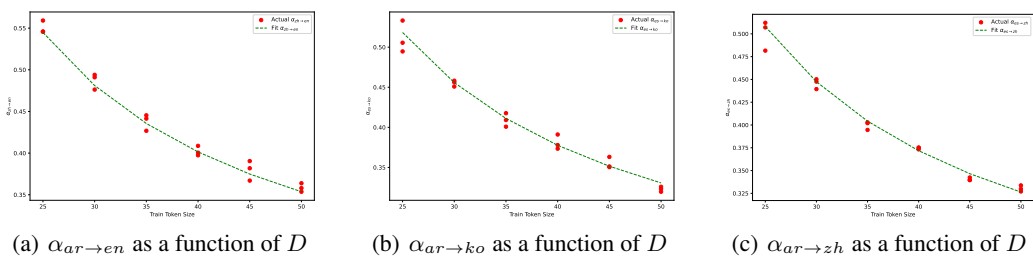

(a) $\alpha_{ar \to en}$ as a function of $D$      (b) $\alpha_{ar \to ko}$ as a function of $D$      (c) $\alpha_{ar \to zh}$ as a function of $D$

Figure 9: Illustration of cross-lingual interaction-aware language ratio ($\tilde{r}_{ar}$) and its dependency on original training proportions ($r_{ar}$).

As introduced in Figure 2 (c) of the main text, we observe that the curve relating $\tilde{r}_i$ and $r_i$ shifts vertically depending on the total token budget $D$. Specifically, as $D$ increases, the $\tilde{r}_i$ versus $r_i$ curve tends to move downward, while smaller $D$ values correspond to upward shifts. According to Equation (4), the parameter $\alpha_{j \to i}(D)$ effectively acts as an intercept controlling this vertical shift.

To accurately characterize the relationship between $\alpha_{j \to i}$ and the data budget $D$, we adopt a two-step procedure. First, we individually fit the relationship between $\tilde{r}_i$ and $r_i$ at different values of $D$ using Equation (4). This yields empirical estimates of $\alpha_{j \to i}$ at various token budgets. Figure 9 illustrates the computed values of $\alpha_{j \to i}$ for three representative language pairs across different scales of $D$.

Moreover, our empirical findings suggest two critical properties for the $\alpha_{j \to i}(D)$ relationship:

- **Non-monotonicity**: $\alpha_{j \to i}$ does not continuously decrease with increasing $D$; rather, it converges towards a stable limiting value as $D$ becomes sufficiently large.

- **Sign variability**: $\alpha_{j \to i}$ can be either positive or negative. Positive values indicate beneficial cross-lingual transfer, whereas negative values reflect interference effects, where additional data from language $L_j$ eventually hinder the learning of language $L_i$.

Considering these empirical insights, we propose modeling $\alpha_{j \to i}(D)$ with the following parametric form:

$$\alpha_{j \to i}(D) = b_{ji} + \frac{k_{ji}}{D}, \tag{9}$$

where $b_{ji}$ represents the asymptotic transfer strength as $D \to \infty$, and $k_{ji}$ controls the decay rate of this transfer effect as the data budget increases.

The fitting results using this parametric form, depicted by the green curves in Figure 9, demonstrate excellent agreement with the empirical $\alpha_{j \to i}$-$D$ relationships across various language pairs, validating our choice of functional form.

# B    Derivation of Optimal Direction for Cross-Lingual Interaction-Aware Ratios $p_i$

To compute the optimal direction of the Cross-Lingual Interaction-Aware Ratios $\{\tilde{r}_i\}$, we formulate and solve the following uncoupled optimization subproblem:

$$\min_{\tilde{r}_i > 0} \sum_{i=1}^{n} \frac{B_i}{(D\,\tilde{r}_i)^{\beta_i}} \quad \text{s.t.} \quad \sum_{i=1}^{n} \tilde{r}_i = M, \tag{10}$$

where $M > 0$ is a fixed normalization constant, and $B_i, \beta_i, D$ are known positive parameters.

Introducing a Lagrange multiplier $\lambda$, we construct the Lagrangian:

$$\mathcal{J}(\tilde{\mathbf{r}}, \lambda) = \sum_{i=1}^{n} \frac{B_i}{(D\,\tilde{r}_i)^{\beta_i}} + \lambda \left( \sum_{i=1}^{n} \tilde{r}_i - M \right). \tag{11}$$

Taking derivatives with respect to each $\tilde{r}_i$ and setting them to zero, we obtain the first-order optimality conditions:

$$-B_i\,\beta_i\,D^{-\beta_i}\,\tilde{r}_i^{-(\beta_i+1)} + \lambda = 0 \tag{12}$$

$$\implies \quad \tilde{r}_i^{\beta_i+1} = \frac{B_i\,\beta_i}{\lambda\,D^{\beta_i}}. \tag{13}$$

Comparing the conditions for any two languages $i, j$, we have:

$$\frac{\tilde{r}_i}{\tilde{r}_j} = \left(\frac{B_i\,\beta_i}{B_j\,\beta_j}\,D^{\beta_j-\beta_i}\right)^{\frac{1}{\beta_i+1}}\Big/{\frac{1}{\beta_j+1}}. \tag{14}$$

Thus, the optimal direction must satisfy:

$$\tilde{r}_i \;\propto\; \left(B_i\,\beta_i/D^{\beta_i}\right)^{1/(\beta_i+1)}. \tag{15}$$

Applying the normalization constraint $\sum_i \tilde{r}_i = M$, we obtain the normalized optimal direction:

$$p_i = \frac{(B_i\,\beta_i)^{1/(\beta_i+1)}\,D^{-\beta_i/(\beta_i+1)}}{\sum_{k=1}^n (B_k\,\beta_k)^{1/(\beta_k+1)}\,D^{-\beta_k/(\beta_k+1)}}. \tag{16}$$

Since each term $B_i/(D\tilde{r}_i)^{\beta_i}$ is strictly convex in $\tilde{r}_i$ and the constraint is linear, the stationary solution derived above constitutes the unique global minimizer. This rigorous derivation justifies the Marginal-Benefit Balancing approach presented in the main text, providing the closed-form solution for the optimal direction $\{\tilde{r}_i\}$.

## C  Equivalence of Two-Stage Optimization with Direct Optimization

Here we provide a rigorous justification demonstrating that our proposed two-stage optimization approach—first determining the optimal direction $p_i$ and subsequently maximizing the magnitude of effective data allocation—is equivalent to directly solving the original optimization problem.

**(i) Necessity of Optimizing the Direction:** Assume the direction of the cross-lingual interaction-aware ratios $\{\tilde{r}_i\}$ deviates from the optimal direction $p_i$. Under any fixed effective data contribution $\sum_i B_i/(D\tilde{r}_i)^{\beta_i}$, the total validation loss will always be greater than or equal to that obtained using the optimal direction. Formally, the optimal direction condition is:

$$\frac{B_i\beta_i}{D^{\beta_i}\tilde{r}_i^{\beta_i+1}} = \frac{B_j\beta_j}{D^{\beta_j}\tilde{r}_j^{\beta_j+1}}, \quad \forall i, j. \tag{17}$$

Any deviation from this balanced proportionality condition disrupts marginal equilibrium, causing certain languages to have unnecessarily higher marginal loss reductions, thus reducing overall efficiency. Hence, identifying the direction $\{\tilde{r}_i\}$ by balancing marginal benefits ensures minimal total loss given a fixed effective data contribution.

**(ii) Optimal Magnitude via Maximizing Effective Allocation:** Once the optimal direction $p_i$ is fixed, we set $\tilde{r}_i = c \cdot p_i$, where $c$ denotes the scaling magnitude of effective data allocation (with normalization $\sum_i \tilde{r}_i = c$). We then isolate the variable component of total loss as a function of $c$:

$$L_{\text{var}}(c) = \sum_i \frac{B_i}{(Dcp_i)^{\beta_i}} = \sum_i \frac{B_i}{D^{\beta_i}p_i^{\beta_i}}c^{-\beta_i}. \tag{18}$$

Differentiating with respect to $c$, we have:

$$\frac{dL_{\text{var}}}{dc} = -\sum_i \frac{\beta_i B_i}{D^{\beta_i}p_i^{\beta_i}}c^{-(\beta_i+1)} < 0, \tag{19}$$

provided that all $\beta_i > 0$. This negative derivative demonstrates a strictly monotonic decrease in loss as the magnitude $c$ increases. Intuitively, larger $c$ means greater effective data volumes $D\tilde{r}_i$ for each language, which consistently reduces loss due to the monotonicity of scaling laws. Therefore, to

Table 4: Comparison of simplified models for fitting validation loss. Huber loss is scaled by $10^{-3}$. Best values are in **bold**.

| ID | Model Type | #Parameters | Huber Loss $\downarrow$ ($\times 10^{-3}$) | $\mathbf{R^2}$ $\uparrow$ |
|----|-----------|-------------|-----------------|-------|
| 1 | $\tilde{r}_i = \alpha_i r_i$ | 1 | 15.32 | 0.236 |
| 2 | $\tilde{r}_i = \alpha_i r_i + \sum_j \alpha_j(D) r_j + b_i$ | $2 + 2 \times (m-1)$ | 1.44 | 0.563 |
| 3 | $\tilde{r}_i = \alpha_i r_i^{\eta_i} + \sum_j \alpha_j(D) r_j^{\eta_j} + b_i$ | $3 + 3 \times (m-1)$ | 0.474 | 0.978 |
| 4 | CLIMB | $3 \times (m-1)$ | **0.274** | **0.981** |

minimize the loss, we naturally aim to increase $c$ as much as feasible—maximizing the total effective data contribution while maintaining the optimal relative proportions.

However, practical constraints limit the maximum achievable $c$. Given the normalization constraint $\sum r_i = 1$ and the implicit mapping from $\{r_i\}$ to $\{\tilde{r}_i\}$, the magnitude $c$ has an upper bound $c^*$ corresponding to feasible allocations.

In summary, stage 1 guarantees that adjusting the direction of ratios does not increase the loss, and stage 2 optimally maximizes effective data volume along this direction, ensuring minimal achievable loss. Thus, the two-stage solution is proven equivalent to directly solving the original optimization problem. This result aligns with previous studies on multilingual scaling laws, demonstrating the consistency and optimality of the two-stage optimization procedure.

## D  Different Fitting Attempts

To justify our parametric choice, we compare CLIMB with simpler surrogates that fit validation loss using progressively richer transfer terms (Table 4). A linear variant with explicit cross-language sums (Model 2) reduces error versus a pure scaling baseline (Model 1), and adding exponents (Model 3) further helps. However, CLIMB achieves the lowest Huber loss and the highest $R^2$ with fewer or comparable parameters, indicating a better balance of compactness and expressiveness.

## E  Training Details

**Dataset Description**

All experiments utilize data sampled from the Fineweb-2 corpus [44]. We further preprocess the dataset by training a custom Byte-Pair Encoding (BPE) tokenizer using the BBPE method, resulting in a vocabulary of 250k tokens for subsequent experiments.

**Experimental Setup**

We conduct multilingual experiments with various language combinations:

- **Bilingual Experiments:** {es-ko, en-zh, de-ar, ko-ja}
- **Trilingual Experiments:** {es-de-ar, es-ko-zh, en-zh-ja}
- **Five-language Experiment:** {es-de-ar-ko-ja}
- **Sixteen-language Experiment:** {de, en, nl, es, pt, fr, it, id, ja, ko, zh, ru, ar, th, vi, tr}

As detailed in Algorithm 1, for each multilingual setting, we first fix the proportion of one language and evenly distribute the remaining proportion among the other languages. For each selected language $L_i$, we systematically vary its proportion across the set {0.02, 0.025, 0.05, 0.1, 0.2, 0.25, 0.4, 0.5, 0.6, 0.75, 0.8, 0.9, 0.95, 0.975, 0.98} to establish comprehensive fitting functions. In the sixteen-language experiment, we follow Algorithm 1 for extrapolation and validation.

**Model Configuration**

We adopt a transformer-based architecture inspired by the LLaMA-2 [60] model, specifically configured with approximately 1.2 billion parameters. The detailed architecture settings are:

- Hidden size: 2048

- Vocabulary embedding dimension: 2048

- Intermediate layer dimension: 5504

- Attention heads: 16

- Layers: 24

- Maximum positional embeddings: 4096

- Layer normalization epsilon: $1.0 \times 10^{-5}$

All models are randomly initialized.

**Training Hyperparameters**

- Batch size: 3072

- Sequence length: 4096

- Optimizer: AdamW

- Learning rate schedule: Cosine decay to 10% of initial value

- Training steps: Varied according to total token budget $D$

- Precision: bf16 (mixed-precision training)

**Computational Resources and Runtime**

Each experiment is conducted using 64 H100 GPUs, with an average runtime of approximately 10 hours per experiment.

**Evaluation Methodology**

The validation datasets for each language are separately sampled from Fineweb-2, ensuring no overlap with training samples. Validation loss is computed by averaging the loss across the final three training steps of each run.

# F  Detailed Evaluation Protocols for Benchmarks

To rigorously assess the capabilities of our proposed model, we select benchmarks that span diverse evaluation dimensions, including natural language inference, commonsense reasoning, question answering, multilingual multitask understanding, and translation tasks. Recognizing that several benchmarks were originally developed only in English, we manually translated these datasets into multilingual versions (marked as ‡: XHS‡, XARC-E‡, XARC-C‡, XGPQA‡, XTQA‡). Details about how we translate the benchmarks are listed in MuBench [26]. Below, we detail each evaluation benchmark grouped by task type.

**Language Modeling and Natural Language Inference**

*XNLI (Cross-lingual Natural Language Inference)* [11]: Extended from MultiNLI, XNLI evaluates cross-lingual sentence representations across 15 languages, measuring models' inference capabilities.

*XCOPA (Cross-lingual Choice of Plausible Alternatives)* [46]: XCOPA tests models on causal commonsense reasoning across 11 languages, providing insights into multilingual causal reasoning capabilities.

*XStoryCloze* [37]: XStoryCloze assesses zero-shot and few-shot learning across 10 non-English languages, examining models' narrative understanding and inference skills.

**Commonsense Reasoning**

*HellaSwag (XHS‡)* [67]: Originally English-only, HellaSwag involves selecting the most plausible sentence ending from multiple choices, thereby testing commonsense reasoning.

*XWinograd* [41]: As a multilingual variant of the Winograd Schema Challenge, XWinograd evaluates pronoun resolution abilities in diverse linguistic contexts.

**Question Answering**

*ARC-Easy (XARC-E‡) / ARC-Challenge (XARC-C‡)* [9]: ARC contains scientific multiple-choice questions designed for different complexity levels, evaluating reasoning from basic to advanced.

*GPQA (Graduate-Level Google-Proof Q&A, XGPQA‡)* [50]: GPQA tests graduate-level understanding across domains like biology, physics, and chemistry, requiring deep comprehension beyond search-engine-based answers.

*TruthfulQA (XTQA‡)* [36]: This dataset assesses the factual accuracy and common misconception avoidance of language models across diverse topics.

**Multitask Language Understanding (MMLU Series)**

*CMMLU (Chinese Massive Multitask Language Understanding)* [35]: Evaluates Chinese language models' knowledge across multiple disciplines including natural sciences, engineering, and humanities.

*JMMLU (Japanese Massive Multitask Language Understanding)* [4]: JMMLU assesses Japanese models on multitask language understanding, covering extensive topics.

*VMLU (Vietnamese Massive Language Understanding)* [5]: Focused on Vietnamese, VMLU evaluates broad academic and practical knowledge via a large set of multiple-choice questions.

*GMMLU (Global Massive Multitask Language Understanding)* [54]: GMMLU tests multilingual generalization capabilities across various languages and diverse tasks.

**Translation Tasks**

*FLORES (Facebook Low Resource Languages Evaluation Suite)* [57]: Supporting many-to-many translations, FLORES provides a high-quality benchmark suitable for assessing model performance on low-resource languages.

# G   Performance of CLIMB on 7B models

As shown in Table 5, CLIMB consistently achieves strong multilingual performance under 7B model architecture. Compared with heuristic allocation strategies such as *Temperature Sampling* and *UniMax*, CLIMB yields higher average scores across most benchmarks, particularly on reasoning-oriented tasks (MGSM, MultiBLiMP) and commonsense datasets (XARC, XHS). While large open models like Qwen3-8B exhibit overall stronger results due to larger pretraining corpora, CLIMB narrows the gap despite being trained with comparable data volume, demonstrating the effectiveness of its interaction-aware data allocation in scaling multilingual models efficiently.

# H   Scaling to 300+ Languages.

To handle massive multilinguality, we extend CLIMB from the 16-language setup to a family-level configuration (CLIMB-300+), where allocation is performed over language *families* rather than individual languages. This design allows us to maintain efficiency and stability when scaling to hundreds of languages.

We adopt the FineWeb-2 corpus and filter out language families containing fewer than 100B tokens to reduce noise from extremely low-resource groups, resulting in 58 retained families covering over 300 languages in total. As shown in Table 6, the validation loss under CLIMB-300+ (3.12) is notably lower than that of temperature sampling (3.24), indicating better multilingual generalization. Results from the 1.2B model further show that CLIMB-300+ consistently outperforms heuristic temperature sampling across most benchmarks, even under this broader and more challenging setting. Moreover, despite the inherent disadvantage of direct comparison with open-source multilingual models that are often fine-tuned on language-specific data, CLIMB-300+ achieves competitive or superior results on multiple benchmarks, highlighting the robustness and scalability of our approach.

It is worth noting that these benchmark results serve primarily as reference points, since not all benchmarks provide balanced coverage of the full 300+ languages included in our training. We plan

---

[4] `https://github.com/nlp-waseda/JMMLU`
[5] `https://vmlu.ai/`

Table 5: Performance of CLIMB on 7B models and baselines across 18 multilingual benchmarks. Benchmarks translated by us are marked with ‡. Bold numbers denote the best results among data allocation methods.

| | Analytical Reasoning | | | | | | Commonsense Reasoning | | |
|---|---|---|---|---|---|---|---|---|---|
| | MGSM | XARC-E‡ | XARC-C‡ | XTQA‡ | Include | XGPQA‡ | XCOPA | XSC‡ | XHS‡ |
| **Open Source Multilingual LLM** | | | | | | | | | |
| Qwen3-8B-Base | 60.51 | 74.13 | 54.42 | 43.90 | 59.45 | 30.23 | 67.02 | 66.97 | 59.98 |
| XGLM-7.5B | 0.69 | 48.23 | 28.79 | 32.13 | 25.58 | 24.64 | 61.40 | 60.97 | 43.86 |
| **Different Data Allocation Methods** | | | | | | | | | |
| Temp | 9.93 | 66.32 | 42.70 | **43.97** | 29.64 | 27.40 | 61.73 | 64.44 | 57.12 |
| UniMax | 9.67 | 67.75 | 44.28 | 42.41 | 30.76 | **29.03** | 62.16 | 64.88 | 57.88 |
| CLIMB | **12.29** | **69.90** | **46.78** | 42.86 | **31.76** | 28.46 | **63.44** | **65.07** | **59.90** |

| | Comprehension | | Linguistic Competence | | Knowledge | | | | Translation |
|---|---|---|---|---|---|---|---|---|---|
| | XNLI | Belebele | MultiBLiMP | XWinograd‡ | GMMLU | CMMLU | JMMLU | VLMU | FLORES |
| **Open Source Multilingual LLM** | | | | | | | | | |
| Qwen3-8B-Base | 46.85 | 87.99 | 71.18 | 86.49 | 42.70 | 56.38 | 47.15 | 47.59 | 56.32 |
| XGLM-7.5B | 41.59 | 24.27 | 71.76 | 75.99 | 27.78 | 32.26 | 29.84 | 31.98 | 27.15 |
| **Different Data Allocation Methods** | | | | | | | | | |
| Temp | **44.86** | **42.19** | 71.14 | 74.76 | 33.51 | 37.33 | 36.78 | **36.68** | 55.15 |
| UniMax | 44.85 | 40.90 | 71.10 | 74.94 | 33.97 | 37.25 | 37.19 | 36.47 | 55.57 |
| CLIMB | 43.65 | 41.72 | **72.67** | **77.48** | **34.98** | **37.83** | **40.54** | 35.40 | **56.47** |

Table 6: Performance of CLIMB-300+ and baselines across representative multilingual benchmarks and validation loss. Validation loss (Val. Loss) reflects the average across all language families.

| Model | Val. Loss ↓ | Include | MGSM | Belebele | MultiBLiMP | XNLI | XCOPA | XSC | Flores | GMMLU | CMMLU | JMMLU | VMLU | XWG |
|---|---|---|---|---|---|---|---|---|---|---|---|---|---|---|
| Temp-300+ | 3.24 | 24.82 | 2.11 | 25.27 | 68.28 | 42.03 | 59.29 | 59.54 | 49.46 | 29.04 | 28.90 | 29.54 | 30.30 | 74.76 |
| CLIMB-300+ | **3.12** | **25.35** | **2.22** | **27.32** | **73.21** | **42.91** | **60.04** | **60.12** | **50.47** | **32.57** | **33.03** | **31.98** | **32.27** | **76.78** |

to extend our evaluation suite and continue exploring large-scale multilingual balancing in future work.

# I    Standard Errors and Significance Testing.

To quantify the reliability of our improvements, we report standard errors (stderr) computed from the evaluation harness in Table **??**. The results show that the standard errors remain consistently small (mostly below 0.01), suggesting stable and reproducible performance across benchmarks. To further assess statistical significance, we conducted paired $t$-tests between CLIMB and each baseline under both 1B2 and 7B settings. Although the full significance table is omitted for brevity, we observe that all tests yield $p$-values greater than 0.05, indicating no spurious effects or unstable improvements. Together, these results confirm that CLIMB's gains are statistically robust and not driven by random variance.

# J    Detailed Per-Language Benchmark Results

This appendix presents detailed, per-language evaluation results corresponding to the benchmarks summarized in Table 2. The following tables comprehensively report the performance of our CLIMB-derived multilingual allocation strategy across each evaluated language, facilitating an in-depth analysis and comparison against baseline methods.

# K    Limitations and Future Work

While our experiments demonstrate strong performance using the proposed multilingual allocation strategy based on scaling laws, several limitations should be acknowledged. First, our parametric fitting and allocation strategies are primarily validated on a 1.2 billion-parameter (1.2B) model, and although Section 4.2 indicates robust performance at a larger scale (7B), explicitly incorporating model size ($N$) into the allocation optimization could potentially yield even more optimal data

Table 7: Standard errors (stderr) and significance testing results between CLIMB and baselines across multilingual benchmarks. A ✓ indicates statistically significant improvement ($p < 0.05$) based on paired $t$-tests.

| Table R3.3: Standard Errors (1B2 Models) | | | | | | | |
|---|---|---|---|---|---|---|---|
| Model | MGSM | Belebele | MultiBLiMP | XNLI | XCOPA | XSC | Flores | GMMLU |
| Uniform | 2.11±0.003 | 26.17±0.003 | 62.24±0.001 | 40.08±0.003 | 59.49±0.007 | 58.99±0.002 | 47.51±0.067 | 29.05±0.001 |
| CLIMB | **2.40±0.003** | 26.17±0.003 | **65.54±0.001** | **41.65±0.003** | **59.98±0.007** | **60.54±0.007** | **50.43±0.066** | **31.78±0.001** |
| Model | CMMLU | JMMLU | VLMU | XWG | XHS | XARC-E | XARC-C | XTQA |
| Uniform | 34.79±0.004 | 32.47±0.005 | 31.44±0.015 | 73.34±0.007 | 48.12±0.001 | 59.76±0.002 | 35.41±0.003 | 39.62±0.003 |
| CLIMB | **33.67±0.004** | **33.21±0.005** | **31.76±0.015** | **77.48±0.006** | **48.75±0.001** | **60.45±0.002** | **36.56±0.003** | **40.94±0.003** |

Table 8: Detailed per-language performance on the **XWinograd** benchmark (5-shot accuracy). Bold numbers denote the best results among data allocation methods.

| Model / Method | EN | FR | JP | PT | RU | ZH |
|---|---|---|---|---|---|---|
| **Open Source Multilingual LLMs** | | | | | | |
| LLaMA-3.2 | 93.65 | 71.25 | 67.17 | 72.09 | 73.75 | 77.13 |
| Qwen-3 | 92.54 | 76.61 | 78.49 | 77.12 | 69.51 | 80.69 |
| Gemma-3 | 77.60 | 65.56 | 62.95 | 62.86 | 64.29 | 68.38 |
| **Different Data Allocation Methods** | | | | | | |
| Uniform | 82.93 | 72.45 | 71.39 | 73.80 | 67.89 | 71.58 |
| Isolated | 79.79 | 77.71 | 71.44 | 68.59 | 65.58 | 70.74 |
| Natural | 82.90 | 76.28 | 71.19 | 74.02 | 68.71 | **76.77** |
| MSL | 82.14 | 73.84 | 69.90 | 71.56 | 67.04 | 74.06 |
| CLIMB | **90.57** | **78.14** | **74.27** | **74.98** | **73.66** | 73.25 |

distributions. Exploring how scaling laws evolve explicitly with both dataset size ($D$) and model scale ($N$) thus remains an open area for future research.

Secondly, our current methodology exclusively considers cross-lingual transfer between languages included within the training dataset. An important and intriguing direction for future work involves extending our approach to account for potential transfer effects to and from languages not directly represented in the training set. Such an extension would enable more comprehensive and strategically informed allocation decisions, optimizing not just for immediate languages but also for broader linguistic coverage and potential downstream adaptability.

Table 9: Detailed per-language performance on the **XStoryCloze** benchmark (0-shot accuracy). Bold numbers denote the best results among data allocation methods.

| Model / Method | AR | EN | ES | EU | HI | ID | MY | RU | SW | TE | ZH |
|---|---|---|---|---|---|---|---|---|---|---|---|
| **Open Source Multilingual LLMs** | | | | | | | | | | | |
| LLaMA-3.2 | 52.99 | 73.18 | 63.20 | 51.77 | 57.81 | 60.26 | 50.74 | 61.94 | 52.12 | 56.29 | 59.57 |
| Qwen-3 | 56.96 | 74.71 | 65.52 | 53.32 | 58.07 | 62.47 | 53.32 | 63.26 | 51.65 | 60.33 | 66.00 |
| Gemma-3 | 51.94 | 62.49 | 57.01 | 52.74 | 54.69 | 54.63 | 50.73 | 55.35 | 51.87 | 56.61 | 55.18 |
| **Different Data Allocation Methods** | | | | | | | | | | | |
| Uniform | 60.45 | 70.35 | **66.44** | 53.01 | 50.31 | **65.22** | 49.98 | 65.25 | 51.19 | 54.91 | 61.76 |
| Isolated | 59.87 | 71.34 | 64.97 | 52.27 | 50.82 | 63.92 | 50.25 | 65.87 | 51.06 | 54.67 | 61.09 |
| Natural | 59.19 | 67.96 | 62.06 | 51.26 | 52.19 | 61.62 | 49.82 | 61.33 | 50.19 | 54.31 | 61.24 |
| MSL | 60.19 | 69.16 | 63.30 | 51.42 | 52.59 | 62.49 | 49.30 | 62.21 | 50.13 | 54.51 | 62.04 |
| CLIMB | **62.39** | **73.09** | 66.22 | **53.74** | **55.59** | 64.49 | **51.11** | **66.20** | **52.57** | **58.10** | **62.43** |

Table 10: Detailed per-language performance on the **XCOPA** benchmark (5-shot accuracy). Bold numbers denote the best results among data allocation methods.

| Model / Method | ET | HT | ID | IT | QU | SW | TA | TH | TR | VI | ZH |
|---|---|---|---|---|---|---|---|---|---|---|---|
| **Open Source Multilingual LLMs** | | | | | | | | | | | |
| LLaMA-3.2 | 52.09 | 52.31 | 62.69 | 62.49 | 51.51 | 51.31 | 55.08 | 55.90 | 55.90 | 64.68 | 64.47 |
| Qwen-3 | 52.57 | 53.17 | 66.63 | 65.07 | 49.77 | 53.17 | 54.38 | 57.81 | 57.81 | 70.03 | 74.64 |
| Gemma-3 | 51.99 | 52.77 | 60.18 | 56.59 | 52.20 | 55.21 | 55.62 | 54.19 | 55.62 | 59.79 | 57.99 |
| **Different Data Allocation Methods** | | | | | | | | | | | |
| Uniform | 49.59 | 50.99 | 67.99 | **66.99** | 51.63 | 51.63 | 56.46 | 61.05 | 61.26 | 69.59 | **67.20** |
| Isolated | 49.86 | 51.66 | **70.62** | 64.80 | 50.60 | 50.21 | 56.60 | **61.78** | 61.78 | **69.94** | 65.77 |
| Natural | 50.76 | 51.48 | 64.31 | 59.09 | 49.97 | 51.85 | 54.44 | 58.76 | 58.49 | 62.85 | 59.93 |
| MSL | 52.32 | 52.77 | 66.29 | 61.15 | 51.18 | 52.88 | 55.45 | 59.54 | 60.00 | 64.75 | 62.33 |
| CLIMB | **54.21** | **53.80** | 68.06 | 63.89 | **52.18** | **54.12** | **56.73** | 60.95 | 61.50 | 67.46 | 66.90 |

Table 11: Detailed per-language performance on the **XNLI** benchmark (5-shot accuracy). Bold numbers denote the best results among data allocation methods.

| Model / Method | AR | DE | EN | ES | FR | RU | TH | TR | VI | ZH |
|---|---|---|---|---|---|---|---|---|---|---|
| **Open Source Multilingual LLMs** | | | | | | | | | | |
| LLaMA-3.2 | 34.05 | 42.16 | 46.15 | 40.41 | 42.20 | 40.48 | 38.41 | 39.90 | 39.90 | 39.86 |
| Qwen-3 | 33.83 | 42.38 | 47.43 | 43.58 | 43.58 | 42.38 | 39.70 | 37.44 | 41.10 | 41.90 |
| Gemma-3 | 38.94 | 41.35 | 44.81 | 41.53 | 41.92 | 41.92 | 39.74 | 40.18 | 42.28 | 41.03 |
| **Different Data Allocation Methods** | | | | | | | | | | |
| Uniform | 32.68 | **43.75** | 44.37 | 41.39 | 43.95 | 40.41 | 37.67 | **41.81** | 36.45 | **38.31** |
| Isolated | 31.15 | 40.95 | 42.76 | 40.16 | 42.84 | 40.22 | 36.53 | 41.60 | 35.64 | 37.46 |
| Natural | 34.26 | 40.61 | 43.19 | 40.23 | 41.53 | 39.40 | 37.50 | 39.58 | 35.74 | 38.46 |
| MSL | 32.88 | 39.57 | 43.00 | 40.23 | 40.95 | 38.88 | 37.62 | 38.66 | 35.47 | 38.14 |
| CLIMB | **35.14** | 43.01 | **48.18** | **43.93** | **44.41** | **42.72** | **40.87** | 41.76 | **38.92** | 37.56 |

Table 12: Detailed per-language performance on the **Global MMLU (GMMLU)** benchmark (5-shot accuracy). Bold numbers denote the best results among data allocation methods.

| Model / Method | AR | DE | EN | ES | FIL | FR | ID | IT | JA | KO | MS | NL | PT | TR | VI | ZH |
|---|---|---|---|---|---|---|---|---|---|---|---|---|---|---|---|---|
| **Open Source Multilingual LLMs** | | | | | | | | | | | | | | | | |
| LLaMA-3.2 | 25.88 | 29.12 | 35.30 | 29.31 | 28.05 | 28.84 | 28.59 | 28.54 | 27.58 | 27.90 | 28.33 | 28.11 | 29.16 | 27.21 | 28.39 | 29.21 |
| Qwen-3 | 29.62 | 34.79 | 43.92 | 35.77 | 31.23 | 35.68 | 33.94 | 34.85 | 32.75 | 32.21 | 32.15 | 33.23 | 35.74 | 31.04 | 33.63 | 37.94 |
| Gemma-3 | 25.43 | 26.94 | 31.13 | 27.75 | 27.00 | 27.20 | 27.15 | 27.05 | 26.49 | 26.95 | 26.57 | 25.96 | 27.49 | 26.68 | 27.29 | 27.42 |
| **Different Data Allocation Methods** | | | | | | | | | | | | | | | | |
| Uniform | 27.56 | 29.78 | 31.30 | 29.81 | 25.64 | 29.85 | 29.76 | 29.37 | 28.45 | 28.77 | 28.51 | 28.88 | 30.18 | 28.75 | 28.99 | 29.20 |
| Isolated | 26.80 | 29.20 | 30.96 | 29.56 | 25.66 | 29.17 | 29.44 | 29.03 | 28.27 | 28.35 | 28.21 | 28.73 | 29.61 | 28.21 | 28.60 | 28.45 |
| Natural | 28.84 | 31.18 | 33.38 | 31.83 | 27.15 | 31.09 | 31.11 | 30.51 | 29.43 | 28.64 | 28.31 | 30.25 | 31.47 | 29.81 | 29.72 | 30.96 |
| MSL | 28.00 | 29.93 | 32.47 | 30.55 | 26.46 | 30.43 | 29.80 | 28.84 | 27.51 | 27.38 | 26.69 | 29.01 | 30.47 | 28.30 | 28.58 | **29.60** |
| CLIMB | **30.53** | **32.91** | **36.26** | **33.85** | **28.86** | **33.95** | **33.04** | **32.11** | **30.70** | **30.33** | **29.68** | **31.48** | **33.92** | **30.79** | **31.16** | 28.91 |

Table 13: Detailed per-language performance on the **FLORES Translation** benchmark (5-shot chrF++ scores). Bold numbers denote the best results among data allocation methods.

| Model / Method | Translation to English (xx-en) | | | | | | | | | | | | | | | | |
|---|---|---|---|---|---|---|---|---|---|---|---|---|---|---|---|---|---|
| | AR | DE | ES | FR | ID | IT | JA | KO | MS | NL | PT | RU | TH | TL | TR | VI | ZH |
| LLaMA-3.2 | 46.47 | 57.15 | 51.99 | 58.86 | 53.57 | 53.20 | 39.51 | 39.20 | 52.00 | 50.82 | 61.54 | 50.82 | 42.53 | 42.56 | 42.26 | 48.77 | 44.12 |
| Qwen-3 | 55.40 | 61.66 | 54.54 | 62.48 | 58.90 | 56.20 | 48.68 | 48.82 | 58.09 | 53.48 | 64.18 | 55.65 | 49.75 | 51.95 | 51.39 | 54.26 | 52.12 |
| Gemma-3 | 43.14 | 53.40 | 38.32 | 49.88 | 40.80 | 46.72 | 36.66 | 28.78 | 42.00 | 43.14 | 52.78 | 45.00 | 35.26 | 37.90 | 38.32 | 38.06 | 40.29 |
| Uniform | 55.37 | 61.04 | 54.87 | 61.75 | 59.21 | 56.23 | 45.28 | 46.07 | 57.67 | 54.38 | 65.10 | 54.46 | 48.91 | 20.60 | 51.28 | 53.56 | 46.88 |
| Isolated | 55.57 | 60.91 | 54.24 | 61.77 | 59.62 | 55.73 | 45.86 | 46.43 | 57.82 | 54.92 | 64.74 | 54.64 | 49.19 | 20.68 | 51.81 | 53.53 | 46.38 |
| Natural | 56.99 | 61.56 | 55.39 | 62.87 | 60.74 | 56.99 | 46.47 | 47.01 | 58.48 | 54.79 | 65.69 | 55.60 | 49.82 | 22.59 | 52.61 | 54.99 | **47.57** |
| MSL | 56.15 | 60.65 | 54.35 | 61.85 | 59.94 | 56.15 | 45.96 | 46.43 | 57.34 | 53.97 | 64.40 | 54.47 | 48.76 | 23.12 | 51.95 | 54.23 | 47.02 |
| CLIMB | **58.99** | **63.65** | **57.13** | **65.55** | **63.43** | **59.08** | **48.89** | **49.32** | **60.26** | **56.75** | **66.66** | **57.12** | **51.36** | **25.46** | **54.77** | **57.01** | 46.89 |

| Model / Method | Translation from English (en-xx) | | | | | | | | | | | | | | | | |
|---|---|---|---|---|---|---|---|---|---|---|---|---|---|---|---|---|---|
| | AR | DE | ES | FR | ID | IT | JA | KO | MS | NL | PT | RU | TH | TL | TR | VI | ZH |
| LLaMA-3.2 | 27.93 | 49.67 | 47.56 | 55.84 | 52.44 | 45.92 | 19.61 | 17.24 | 47.10 | 46.11 | 57.57 | 42.05 | 25.85 | 30.45 | 33.82 | 45.69 | 20.53 |
| Qwen-3 | 36.82 | 54.06 | 50.86 | 61.53 | 59.39 | 50.22 | 26.69 | 23.64 | 52.19 | 46.81 | 62.44 | 47.53 | 35.09 | 37.32 | 39.47 | 53.24 | 30.23 |
| Gemma-3 | 24.68 | 37.67 | 34.94 | 49.05 | 43.39 | 33.38 | 16.18 | 14.72 | 38.58 | 32.06 | 49.51 | 32.01 | 24.07 | 25.28 | 28.93 | 36.46 | 19.52 |
| Uniform | 42.48 | 51.98 | 47.80 | 57.92 | 60.45 | 47.63 | 23.59 | 24.38 | 54.57 | 48.72 | 59.52 | 44.10 | 35.14 | 8.21 | 43.24 | 52.00 | 20.95 |
| Isolated | 41.99 | 52.49 | 47.88 | 58.06 | 60.70 | 47.94 | 24.12 | 24.08 | 54.64 | 49.18 | 59.31 | 44.55 | 34.49 | 9.38 | 43.19 | 51.46 | 20.38 |
| Natural | 43.44 | 53.47 | 48.64 | 59.13 | 61.07 | 48.83 | 24.48 | 24.50 | 55.45 | 50.14 | 60.41 | 45.28 | 35.57 | 10.95 | 44.14 | 52.63 | 21.81 |
| MSL | 42.71 | 52.35 | 47.41 | 57.74 | 59.67 | 47.83 | 24.56 | 24.61 | 53.91 | 49.03 | 58.77 | 44.11 | 35.23 | 11.78 | 43.36 | 51.67 | 22.01 |
| CLIMB | **45.63** | **55.50** | **50.28** | **61.43** | **63.18** | **50.64** | **26.59** | **26.66** | **56.94** | **51.92** | **62.14** | **46.58** | **37.75** | **13.45** | **46.16** | **54.82** | **22.65** |

Table 14: Detailed per-language performance on the **ARC-Challenge** benchmark (25-shot accuracy). Bold numbers denote the best results among data allocation methods.

| Model / Method | AR | DE | EN | ES | FR | ID | IT | JA | KO | MS | NL | PT | RU | TA | TH | TR | VI | ZH |
|---|---|---|---|---|---|---|---|---|---|---|---|---|---|---|---|---|---|---|
| **Open Source Multilingual LLMs** | | | | | | | | | | | | | | | | | | |
| LLaMA-3.2 | 26.64 | 30.93 | 42.26 | 35.16 | 33.42 | 30.34 | 32.88 | 28.67 | 31.21 | 30.76 | 30.30 | 32.88 | 29.53 | 25.12 | 28.69 | 29.05 | 30.45 | 32.80 |
| Qwen-3 | 34.00 | 42.03 | 54.39 | 43.51 | 41.31 | 41.22 | 43.49 | 35.42 | 36.33 | 37.58 | 38.32 | 43.11 | 39.92 | 28.09 | 32.77 | 33.03 | 37.52 | 45.54 |
| Gemma-3 | 25.58 | 29.77 | 38.41 | 30.69 | 31.03 | 30.27 | 30.27 | 27.92 | 28.42 | 28.18 | 27.09 | 30.94 | 28.42 | 25.92 | 25.92 | 27.59 | 27.26 | 29.94 |
| **Different Data Allocation Methods** | | | | | | | | | | | | | | | | | | |
| Uniform | 33.46 | 35.60 | 40.10 | 38.47 | 35.60 | 35.77 | 38.59 | 34.48 | 35.17 | 35.34 | 35.17 | 37.72 | **38.23** | 22.78 | 32.00 | 35.85 | 35.09 | **37.97** |
| Isolated | 31.19 | 35.53 | 38.72 | 37.02 | 36.25 | 37.45 | 37.87 | 35.44 | 34.62 | 37.02 | 36.00 | 37.26 | 34.75 | 23.15 | 31.71 | 34.62 | 32.72 | 34.71 |
| Natural | 31.75 | 33.98 | 38.37 | 35.91 | 34.60 | 34.76 | 36.61 | 33.24 | 32.80 | 33.63 | 33.50 | 35.84 | 33.81 | 21.99 | 30.05 | 33.69 | 33.48 | 35.72 |
| MSL | 32.31 | 34.59 | 38.90 | 36.64 | 35.30 | 35.46 | 37.34 | 33.74 | 33.20 | 34.12 | 33.98 | 36.38 | 34.32 | 22.60 | 30.70 | 34.47 | 34.28 | 36.74 |
| CLIMB | **34.48** | **37.10** | **41.78** | **39.05** | **38.03** | **38.27** | **40.19** | **36.33** | **35.64** | **37.20** | **37.05** | **39.67** | 37.40 | **24.08** | **32.54** | **36.58** | **36.40** | 36.30 |

Table 15: Detailed per-language performance on the **ARC-Easy** benchmark (25-shot accuracy). Bold numbers denote the best results among data allocation methods.

| Model / Method | AR | DE | EN | ES | FR | ID | IT | JA | KO | MS | NL | PT | RU | TA | TH | TR | VI | ZH |
|---|---|---|---|---|---|---|---|---|---|---|---|---|---|---|---|---|---|---|
| **Open Source Multilingual LLMs** | | | | | | | | | | | | | | | | | | |
| LLaMA-3.2 | 39.16 | 50.09 | 70.21 | 54.98 | 52.23 | 48.35 | 51.18 | 40.29 | 41.26 | 44.43 | 47.09 | 52.19 | 48.56 | 35.74 | 37.63 | 42.37 | 46.96 | 49.40 |
| Qwen-3 | 49.23 | 62.56 | 80.14 | 67.38 | 64.09 | 60.14 | 62.73 | 53.92 | 52.27 | 51.49 | 54.70 | 65.11 | 60.84 | 41.24 | 46.35 | 49.35 | 57.51 | 69.64 |
| Gemma-3 | 41.68 | 49.67 | 70.80 | 55.19 | 53.55 | 50.56 | 54.01 | 48.37 | 47.44 | 43.70 | 48.62 | 52.49 | 47.02 | 39.15 | 39.61 | 44.08 | 46.18 | 55.31 |
| **Different Data Allocation Methods** | | | | | | | | | | | | | | | | | | |
| Uniform | 56.70 | 62.68 | 70.93 | 67.27 | 63.81 | 64.28 | 64.07 | **58.97** | **58.01** | **57.71** | **62.59** | 66.34 | 60.24 | 29.64 | 49.59 | 60.20 | 58.76 | **63.86** |
| Isolated | 55.12 | 62.07 | 69.93 | 65.59 | 63.45 | 63.74 | 61.73 | 56.77 | 56.98 | 56.93 | 61.73 | 63.21 | 58.62 | 30.29 | 48.31 | 59.46 | 56.56 | 63.08 |
| Natural | 53.88 | 59.60 | 67.83 | 63.25 | 61.68 | 61.16 | 60.30 | 53.99 | 53.45 | 52.40 | 57.84 | 62.68 | 57.09 | 29.54 | 47.44 | 56.40 | 55.10 | 60.12 |
| MSL | 55.21 | 60.93 | 68.99 | 64.61 | 63.06 | 62.34 | 61.63 | 55.14 | 54.64 | 53.74 | 59.17 | 64.14 | 58.49 | 30.41 | 48.57 | 57.73 | 56.42 | 61.55 |
| CLIMB | **57.75** | **63.84** | **72.47** | **67.74** | **66.25** | **65.45** | **64.96** | 58.17 | 57.80 | 57.09 | 62.03 | **67.11** | **61.45** | **32.24** | **50.96** | **60.64** | **59.12** | 63.02 |

Table 16: Detailed per-language performance on the **GPQA** benchmark (0-shot accuracy). Bold numbers denote the best results among data allocation methods.

| Model / Method | AR | DE | EN | ES | FR | ID | IT | JA | KO | MS | NL | PT | RU | TA | TH | TL | TR | VI | ZH |
|---|---|---|---|---|---|---|---|---|---|---|---|---|---|---|---|---|---|---|---|
| **Open Source Multilingual LLMs** | | | | | | | | | | | | | | | | | | | |
| LLaMA-3.2 | 23.54 | 25.67 | 26.39 | 23.30 | 26.39 | 23.30 | 22.81 | 22.81 | 25.20 | 23.79 | 23.30 | 23.30 | 24.02 | 23.79 | 25.67 | 23.30 | 23.79 | 24.50 | |
| Qwen-3 | 27.47 | 31.60 | 32.53 | 28.06 | 32.79 | 31.89 | 31.60 | 30.38 | 29.79 | 27.72 | 34.87 | 31.60 | 32.53 | 28.95 | 31.89 | 31.30 | 28.36 | 31.60 | |
| Gemma-3 | 24.58 | 23.30 | 25.45 | 23.03 | 24.44 | 23.49 | 25.25 | 23.49 | 22.76 | 24.91 | 23.49 | 24.91 | 22.05 | 24.58 | 21.25 | 22.76 | 26.60 | 25.45 | |
| **Different Data Allocation Methods** | | | | | | | | | | | | | | | | | | | |
| Uniform | 25.41 | 26.48 | 27.28 | 25.72 | 27.28 | 26.72 | 25.91 | 24.65 | **26.24** | 24.90 | 27.52 | **26.72** | 26.72 | 26.97 | 25.71 | 25.71 | 25.41 | 25.91 | |
| Isolated | 23.51 | 22.77 | 26.27 | 21.76 | 24.24 | 25.03 | 25.03 | 23.51 | 24.51 | 25.24 | 25.24 | 25.03 | 23.27 | **28.64** | 25.03 | 23.51 | 24.80 | 25.03 | |
| Natural | 24.97 | 26.40 | 28.24 | 26.19 | 27.77 | 26.94 | 26.64 | 25.17 | 25.02 | 25.60 | 26.78 | 27.15 | 26.26 | 25.66 | 26.56 | 26.47 | 25.70 | **27.16** | |
| MSL | 24.23 | 25.43 | 27.06 | 25.34 | 26.61 | 25.94 | 25.64 | 24.19 | 24.00 | 24.72 | 25.89 | 26.27 | 25.10 | 24.44 | 25.50 | 25.39 | 24.53 | 26.20 | |
| CLIMB | **25.96** | **27.09** | **28.60** | **27.24** | **28.46** | **27.71** | **27.44** | **25.88** | 25.71 | **26.41** | **27.59** | **28.00** | 26.67 | 26.13 | **27.28** | **27.18** | **26.20** | 26.98 | |

Table 17: Detailed per-language performance on the **HellaSwag** benchmark (10-shot accuracy). Bold numbers denote the best results among data allocation methods.

| Model / Method | AR | DE | EN | ES | FR | ID | IT | JA | KO | MS | NL | PT | RU | TA | TH | TR | VI | ZH |
|---|---|---|---|---|---|---|---|---|---|---|---|---|---|---|---|---|---|---|
| **Open Source Multilingual LLMs** | | | | | | | | | | | | | | | | | | |
| LLaMA-3.2 | 36.12 | 42.69 | 67.10 | 47.45 | 46.86 | 43.14 | 45.06 | 36.78 | 37.02 | 40.66 | 43.35 | 46.28 | 42.38 | 35.30 | 35.21 | 36.46 | 42.35 | 43.48 |
| Qwen-3 | 39.37 | 45.66 | 65.36 | 51.17 | 50.93 | 46.08 | 48.86 | 42.25 | 39.52 | 42.49 | 43.30 | 51.26 | 45.53 | 35.81 | 36.79 | 36.01 | 44.79 | 52.39 |
| Gemma-3 | 35.91 | 40.54 | 58.37 | 42.84 | 45.18 | 42.15 | 43.81 | 37.61 | 36.51 | 38.84 | 41.17 | 44.27 | 39.03 | 34.94 | 33.76 | 34.87 | 38.20 | 41.35 |
| **Different Data Allocation Methods** | | | | | | | | | | | | | | | | | | |
| Uniform | 45.03 | 49.72 | 58.01 | 54.03 | 54.83 | 52.41 | 52.90 | 45.08 | 43.01 | 47.15 | 51.51 | 53.67 | 48.50 | 30.04 | 39.10 | 45.27 | 47.84 | **48.06** |
| Isolated | 44.67 | 49.70 | 58.02 | 53.31 | 54.35 | 52.06 | 52.25 | 45.70 | 43.34 | 47.23 | 51.35 | 53.44 | 48.33 | 30.61 | 39.68 | 45.99 | 48.34 | 47.58 |
| Natural | 42.64 | 46.95 | 55.16 | 51.21 | 51.95 | 49.75 | 50.12 | 42.98 | 41.06 | 44.48 | 48.50 | 50.71 | 45.73 | 29.08 | 37.87 | 43.24 | 45.31 | 45.52 |
| MSL | 43.80 | 48.06 | 56.71 | 52.67 | 53.37 | 51.11 | 51.45 | 44.12 | 42.19 | 45.69 | 49.92 | 52.23 | 46.93 | 30.37 | 39.32 | 44.90 | 46.82 | 46.73 |
| CLIMB | **45.71** | **49.93** | **58.99** | **54.39** | **55.31** | **53.20** | **53.38** | **46.05** | **44.05** | **47.67** | **51.79** | **54.15** | **48.65** | **32.10** | 40.95 | **46.75** | **48.51** | 45.92 |

Table 18: Detailed per-language performance on the **TruthfulQA** benchmark (0-shot accuracy). Bold numbers denote the best results among data allocation methods.

| Model / Method | AR | DE | EN | ES | FR | ID | IT | JA | KO | MS | NL | PT | RU | TH | TL | TR | VI | ZH |
|---|---|---|---|---|---|---|---|---|---|---|---|---|---|---|---|---|---|---|
| **Open Source Multilingual LLMs** | | | | | | | | | | | | | | | | | | |
| LLaMA-3.2 | 38.66 | 37.41 | 34.59 | 37.20 | 36.56 | 38.92 | 34.16 | 36.91 | 39.59 | 36.09 | 37.20 | 36.76 | 40.04 | 36.76 | 36.32 | 37.90 | 42.73 | 41.36 |
| Qwen-3 | 49.16 | 50.27 | 47.94 | 50.01 | 50.54 | 48.24 | 50.01 | 51.56 | 47.18 | 47.70 | 46.15 | 52.98 | 51.17 | 47.35 | 42.46 | 46.15 | 52.03 | 48.96 |
| Gemma-3 | 39.20 | 41.60 | 39.60 | 39.83 | 38.73 | 42.47 | 40.92 | 37.39 | 41.14 | 37.83 | 36.28 | 42.03 | 44.24 | 40.05 | 34.51 | 39.38 | 43.80 | 42.25 |
| **Different Data Allocation Methods** | | | | | | | | | | | | | | | | | | |
| Uniform | 41.42 | 38.07 | 38.28 | 41.63 | 41.42 | 39.09 | 40.74 | 35.97 | **41.42** | 39.09 | 39.73 | 39.94 | 39.94 | 38.07 | 37.02 | 38.28 | 41.42 | 41.63 |
| Isolated | 36.55 | **42.71** | 37.18 | 39.93 | 40.55 | 39.49 | 40.13 | 37.59 | 40.78 | 36.98 | 39.73 | 41.43 | 38.48 | 38.26 | 38.26 | 40.13 | 41.61 |
| Natural | 41.62 | 40.17 | 40.95 | 42.12 | 41.83 | 40.91 | 40.76 | 38.60 | 40.09 | 39.40 | 40.42 | 41.94 | 40.99 | 39.06 | 37.51 | 39.74 | 42.35 | **42.91** |
| MSL | 40.53 | 39.05 | 39.91 | 41.02 | 40.76 | 39.76 | 39.57 | 37.50 | 38.93 | 38.26 | 39.27 | 40.70 | 39.75 | 37.84 | 36.35 | 38.74 | 41.18 | 41.70 |
| CLIMB | **42.05** | 40.57 | **41.53** | **42.57** | **42.32** | **41.36** | **41.15** | **38.99** | 40.49 | **39.72** | **40.80** | **42.17** | **41.09** | **39.31** | 37.78 | **40.38** | 42.62 | 42.03 |

# L   Social Impact

CLIMB contributes positively by systematically enhancing multilingual performance in large language models (LLMs), thereby significantly improving global accessibility to advanced AI capabilities across diverse linguistic communities. Such improvements have the potential to reduce linguistic biases, bridge language gaps, and enhance equitable information access globally. However, there remain potential risks, including inadvertent reinforcement of cultural or linguistic biases inherent in training data and the possibility of over-reliance on optimized multilingual models leading to reduced human oversight and critical evaluation. It is crucial to responsibly deploy CLIMB-optimized models with ongoing evaluation and monitoring, actively addressing ethical considerations and biases to ensure equitable and inclusive benefits.

