# OpenReview forum: "Exploring Polyglot Harmony: On Multilingual Data Allocation for  Large Language Models Pretraining"
_NeurIPS.cc/2025/Conference — NeurIPS 2025 poster_

### Official Review · Reviewer_ybjW · 2025-06-20

**Clarity:** 4
**Significance:** 3
**Originality:** 4
**Rating:** 5
**Confidence:** 4

**Summary:**

This paper proposed a new method to optimize the proportion of training data allocated for different languages in a multilingual language model. They propose a cross-lingual interaction-aware ratio which, based on LLM scaling laws, computes the effective language proportions considering interactions with other languages. They then propose a parametric model of this ratio, which can be fitted based on LLM experiments with varied language proportions. The last stage of their method is to determine, based on the aforementioned fitted model, the optimal ratios for a set of languages and a specified token budget. This non-convex optimization problem is solved through a 2-step process that first finds the direction of the language ratio vector and then finds its magnitude by maximizing a constrained effective allocation problem. The authors test their framework on validation loss prediction, where it outperforms baselines. Using their method to compute optimal ratios, they also train a language model and evaluate it on downstream tasks, where it outperforms models trained with baseline ratio allocators and performs competitively with existing LLMs trained on much larger datasets.

**Questions:**

1. Would you be able to address Weakness #1 by including additional loss prediction models in the final version of your paper? I do think it would strengthen your claims.

2. For a long time the standard approach to multilingual data allocation was upsampling under-represented languages (e.g. as in XLM-R). Is this included in any of your baselines? It’s not explicitly stated, but perhaps it’s used by one of the baselines you compare against.

**Ethical Concerns:**

["NO or VERY MINOR ethics concerns only"]

**Final Justification:**

The authors responded to my main queries and included new results to address my concerns. I was happy to recommend accept initially and I stand by this recommendation.

**Limitations:**

yes

**Quality:**

3

**Strengths And Weaknesses:**

**Strengths:**

1. The contributions of the paper are original and interesting. Well done to the authors! It’s always refreshing to find a rigorous solution to a problem, in this cases language ratio allocation, which is usually approached with simple heuristics. The authors frame the task as an optimization problem, so their contributions align well with NeurIPS.

2. The proposed method is well motivated and clearly formulated. By drawing on existing work on scaling laws, the authors present a logical framework for thinking about language ratio allocation as an optimization problem. Each stage of their framework is presented clearly (in section 2).

3. The validation loss prediction results are strong. The proposed method comfortably outperforms baselines, highlighting the predictive power of the framework. Similarly, the downstream performance is strong, highlighting the practical value of the framework.

**Weaknesses:**

1. The validation loss prediction results would be more convincing if more baselines were included. The biggest question I have after reading the paper is: is it really necessary to go through this complicated, multi-stage model to obtain this kind of predictive power? Aren’t there simpler parametric models that we can fit with language ratios as input and validation loss as the output to be predicted? I understand that there is limited existing work along this line, but the authors would be able to create simple baselines themselves by fitting, e.g. regression models, on the data collected in their experiments.

2. The paper would be improved by having more details about the different models trained by the authors for their experiments. For example, in 3.1 they mention “we vary the token budgets from 5 billion to 100 billion tokens”, but don’t specify in what intervals or how many models are trained. I know much of these details are included in the appendix, but it would be better to include in the main paper body if possible, since the claims of the paper depend heavily on these experiments.

3. This is a small note, but I don’t think the authors should claim “state-of-the-art” performance (as they do in the abstract, introduction), since their models are not state-of-the-art - they are better than comparable baselines, but not better than the best LLMs available.

---

> ### Author Rebuttal · Authors · 2025-07-31
>
> Thank you for your detailed and positive review.
>
> ---
>
> ### Q4.1: Can more experimental details be moved from the appendix?
>
> Thank you for the suggestion. We will revise the camera-ready version to **move key experimental details from the appendix into the main text**, particularly information on the **number of models trained**, **token budget intervals**, and **training configurations**. We agree that this will improve clarity and self-containment.
>
> ---
>
> ### Q4.2: Is it appropriate to claim “state-of-the-art” performance?
>
> Thank you for pointing this out. We agree that the term may be misleading given the limited training budget and data quality. We will **revise or remove** the phrase “state-of-the-art” from the abstract and main text. Instead, we will emphasize that CLIMB achieves **competitive results under constrained training and compute conditions**.
>
> ---
> ### Q4.3: Why not use simpler models to fit validation loss?
>
> Thank you for this valuable suggestion. We fully agree that it is important to justify the complexity of our parametric model. To this end, we implemented and compared several **simpler baselines**, as summarized below:
>
> | ID | Model Type | #Parameters | Huber Loss ↓ $(\times 10^{-3})$ | R² ↑ |
> |----|------------|-------------|--------------|------|
> | 1  | $\tilde{r}_i = \alpha_i r_i$| 1 | 15.32 | 0.236 |
> | 2  | $\tilde{r}_i =\alpha_i r_i + \sum \alpha_j(D) r_j + b_i$ | $2 + 2 \times (m - 1)$ | 1.44 | 0.563 |
> | 3  | $\tilde{r}_i =\alpha_i r_i^{\eta_i} + \sum \alpha_j(D) r_j^{\eta_j} + b_i$ | $3 + 3 \times (m - 1)$ | 0.474 | 0.978 |
> | 4  | CLIMB | $3 \times (m - 1)$ | **0.274** | **0.981** |
>
> - **Model 2** is a linear regression model with explicit transfer terms. It serves as the simplest meaningful baseline beyond uniform allocation.
> - Despite being lightweight, **simpler models (ID 1–3)** yield significantly higher error or lower R² scores.
> - Even **Model 3**, which introduces exponentiation and has **more parameters than our model**, still underperforms our proposed formulation (Model 4).
>
> This demonstrates that our final parametric form is not only compact but also more expressive and better aligned with the empirical patterns in the data. We will include this comparison in the appendix of the final version to make the justification more transparent.
>
>
> ---
>
> ### Q4.4: Did you include upsampling-style baselines like XLM-R?
>
> Thank you for raising this point. We included two strong and widely-used upsampling-style baselines:
>
> - **Temperature sampling (Temp)**, which smooths the token distribution to favor low-resource languages.
> - **UniMax** (suggested by Reviewer 1T2b), which boosts the contribution of the most frequent language in each batch.
>
> We evaluated both baselines under **identical 1B2 and 7B training setups**, and report results in Tables R4.2 and R4.3. Across all multilingual benchmarks, **CLIMB consistently outperforms both Temp and UniMax**, demonstrating its superior effectiveness under comparable conditions.
>
> We will include these results in the revised version for completeness.
>
>
> **Table R4.2** 1B2 model Performance on new baselines and new benchmarks.
> | 1B2 | include | mgsm | belebele | multiblimp | XNLI  | XCOPA | XSC   | Flores | GMMLU | CMMLU | JMMLU | VLMU  | XWG   | XHS   | XARC-E | XARC-C | XTQA  |
> |--------|---------|------|----------|-------------|-------|--------|--------|--------|--------|--------|--------|--------|--------|--------|---------|---------|--------|
> | Temp   | 24.82   | 2.11 | 25.27    | 60.75       | 41.03 | 58.69  | 59.54  | 47.46  | 29.04 | 33.03 | 31.98 | 30.27 | 74.76 | 47.56 | 59.31  | 34.93  | 39.38 |
> | UniMax | 25.12   | 2.07 | 23.30    | 62.24       | 40.88 | 59.35  | 58.99  | 49.12  | 31.27 | **33.75** | 32.26 | 31.16 | 74.94 | 48.22 | 59.39  | 35.78  | 39.67 |
> | CLIMB  | **25.92**   | **2.40** | **26.17**    | **65.54**       | **41.65** | **59.98**  | **60.54**  | **50.43**  | **31.78** | 33.67 | **33.21** | **31.76** | **77.48** | **48.75** | **60.45**  | **36.56**  | **40.94** |
>
> **Table R4.3** 7B model Performance on new baselines and new benchmarks.
> | 7B | include | mgsm | belebele | multiblimp | XNLI  | XCOPA | XSC   | Flores | GMMLU | CMMLU | JMMLU | VLMU  | XWG   | XHS   | XARC-E | XARC-C | XTQA  |
> |--------|---------|------|----------|-------------|-------|--------|--------|--------|--------|--------|--------|--------|--------|--------|---------|---------|--------|
> | Temp   | 29.64   | 9.93 | **42.19**    | 71.14       | **44.86** | 61.73  | 64.44  | 55.15  | 33.51 | 37.33 | 36.78 | **36.68** | 82.72 | 57.12 | 66.32  | 42.70  | **43.97** |
> | UniMax | 30.76   | 9.67 | 40.90    | 71.10       | 44.85 | 62.16  | 64.88  | 55.57  | 33.97 | 37.25 | 37.19 | 36.47 | 82.40 | 57.88 | 67.75  | 44.28  | 42.41 |
> | CLIMB  | **31.76**   | 12.29| 41.72    | **72.67**       | 43.65 | **63.44**  | **65.07**  | **56.47**  | **34.98** | **37.83** | **40.54** | 35.40 | **84.02** | **59.90** | **69.90**  | **46.78**  | 42.86 |

---

### Official Review · Reviewer_iYcL · 2025-07-02

**Clarity:** 3
**Significance:** 3
**Originality:** 3
**Rating:** 4
**Confidence:** 5

**Summary:**

The paper proposes a method for estimating training data ratios for different languages, taking into account language-specific interactions. The authors train several 1B and 7B models using different methods to determine language data proportions and argue that their proposed method offers the strongest results.

**Questions:**

* Appendix D (L70) implies that all models were trained using the same custom tokenizer. Is that correct? If so, how much of the tokenizer represents the different languages used in these experiments? How does compression differ across the languages? How might this impact the results of these experiments?
* For GlobalMMLU, did you use the culturally specific or culturally agnostic subset (or both)?
* Regarding the results in Appendix F, why do you think CLIMB underperforms existing models so consistently in GMMLU, GPQA, and TruthfulQA? Qwen beats it on almost every language. These are the more recently developed benchmarks and the ones that are most relevant in current research.

Misc.
* The lack of bolding in Table 8 makes it hard to readNo bolding on Table 8? Makes it hard to read. But I think this way of organizing the models together is best.
* Weird spacing in “Evaluation Benchmarks” (L275-281)

**Ethical Concerns:**

["NO or VERY MINOR ethics concerns only"]

**Final Justification:**

The authors have mostly addressed my questions and concerns. I still feel that using the same set of models to motivate the experiments as to conduct the experiments is somewhat circular. However, I understand that training so many models from scratch is expensive and time-consuming. As long as the authors clarify this in the final version, I think this is acceptable.

**Limitations:**

No, this paper does not discuss the limitations of the evaluation method, in particular the potential impact of using machine-translated benchmarks. The authors state that they plan to release these benchmarks, however, without human validation, they may be adding low-quality benchmarks into the ecosystem, which may negatively impact work on the languages this paper works on.

**Quality:**

3

**Strengths And Weaknesses:**

**Strengths**:
* The paper tackles an interesting problem, and a solution to it could help train more powerful and equitable models.
* The paper is generally well written and clear.
* The cross-lingual interaction-aware language ratio method seems generally sound and could be useful for other research. The results in Table 1 are compelling.


**Weaknesses**:
* The motivation for these experiments is circular. The authors state that the empirical observations in Section 2.2.1 motivate the following experiments (L144). But are these the same models that are used in the subsequent experiments? Either way, the authors could motivate the four points in Section 2.2.1 with evidence from previous work, with which the authors did not engage very deeply.
[1] shows that the efficacy of crosslingual transfer begins to decrease if the amount of added data from other languages too greatly exceeds the amount of data in the target language (“Dependency on absolute language proportion”, L130-132). [1] also shows that the similarity (specifically in terms of syntactic typology) between the target language and the languages for which there is added data significantly impacts crosslingual transfer (“Dependency on co-training languages”, L133-135). “Dependency on model language counts” (L136-139) is predicted by [2] (e.g. blue bars in Fig. 4). As the number of languages a model is trained on increases, the model will hit capacity limitations.
All of this work falls under the topic of “the curse of multilinguality”, which is not mentioned in the paper.
* The existing multilingual models that the authors compare their models to are all closed-data models. None of the models’ technical reports even state what proportion of the training data is in each of the different languages. Comparing against a model that is either open data or reports training data proportions, e.g. XGLM [3], would make for more meaningful comparisons. XGLM is available in both 1B and 7B sizes.
* The reporting of results in Table 2 is somewhat misleading. The bolded values do not represent the highest values, but only the highest values among the models trained by the authors. The same is true for the appendices (with the exception of Table 8, which allows for easier comparison, though does not contain any bolding). The authors should indicate overall best performance for each task and/or clarify the bolding strategy in the caption for the table.
* Relatedly, the differences across the different models are extremely small in some cases, and it is unclear whether differences would be statistically significant. The authors did not include any standard error. The justification for this decision (L700-701) is insufficient, as it is not necessary to train different random seeds to get standard error. Running evaluations through tools like the LM Evaluation Harness [4] automatically generate standard error, which helps the reader to identify meaningful differences.
This seems especially critical for comparisons between Uniform and CLIMB models, where scores are often very similar. Whether or not these differences are significant impacts the fundamental argument in this paper.
* The results are presented differently for the 7B (Fig. 6) and 1B (Table 2) models. The 7B models aren’t compared to any existing multilingual models. I found unclear which model the results in Table 2 were for. The appendices fail to report any evaluations for the 7B models as far as I can tell. The only comparisons that can be made are with the models trained by the authors (though, again, no standard error is reported, so it is unclear whether differences in performance are meaningful). The 7B results are not as convincing, therefore, as the 1B models. If CLIMB is not as effective at the 7B scale, this would significantly impact the argument of the paper.
* The use of ‘state-of-the-art’ (L287) seems inappropriate since the authors do not show that CLIMB significantly outperforms the existing models. In several cases, Qwen numerically outperforms CLIMB (1B). There are no comparisons at all on the 7B scale.
* The paper is missing some critical details about the translated benchmarks. The only information about how the benchmarks were translated is one sentence in Appendix E, stating that the authors use “GPT-based translation” (L115-116). The authors should at the bare minimum report what kind of translation system was used to translate the benchmarks. Additionally, the authors should add discussion about how well those systems perform on the different languages that you translate the benchmarks into and whether any additional checking was done to verify translation quality.
The authors do not explain why they translate benchmarks instead of using existing multilingual translations of those benchmarks (HellaSwag [5, 6]; GPQA [7, 8]; TruthfulQA) [9]) or use existing (not machine translated) benchmarks (Belebele for reading comprehension in 122 languages [10], MultiBLiMP for grammatical generalization for 101 languages [11]). There are at least two multilingual mathematical reasoning benchmarks MGSM [12] and MCLM [13].
The evaluation results would be more convincing if the authors remove their machine translated benchmarks and use existing (ideally, at least human-validated machine translated) benchmarks.

[1] https://aclanthology.org/2024.emnlp-main.236/

[2] https://aclanthology.org/2020.acl-main.747

[3] https://arxiv.org/pdf/2112.10668

[4] https://github.com/EleutherAI/lm-evaluation-harness/tree/main

[5] https://arxiv.org/abs/2410.08928

[6] https://arxiv.org/abs/2307.16039

[7] https://arxiv.org/abs/2504.11833?

[8] https://arxiv.org/abs/2502.07346

[9] https://arxiv.org/abs/2406.14434?

[10] https://aclanthology.org/2024.acl-long.44/

[11] https://arxiv.org/abs/2504.02768?

[12] https://openreview.net/forum?id=fR3wGCk-IXp

[13] https://arxiv.org/abs/2502.17407?

---

> ### Author Rebuttal · Authors · 2025-07-31
>
> Thank you for your detailed and thoughtful review. We have carefully addressed your comments and made concrete improvements accordingly. **If there are any remaining concerns, we would be more than willing to engage further and are genuinely eager for a reconsideration of the score.**
>
> ---
> ### Q3.1: Are the experiments in Section 2.2.1 circular in motivation?
>
> Thank you for raising this important concern. We confirm that there is **no circularity** in our experimental design. Our methodology follows the standard approach in scaling law research. Specifically, we trained over **500 different 1B2 models** with diverse, randomly sampled language ratios and data scales across 16 languages. These controlled experiments allow us to **quantitatively measure cross-lingual interactions**, rather than relying on assumptions. The patterns shown in **Figure 2** are derived purely from these randomized runs.
>
> We then fit a **scaling law function (CLIMB)** to these empirical results, capturing how validation loss depends on both data scale and multilingual composition. As shown in **Table 1**, the fitted function generalizes well to unseen settings, validating its accuracy. Based on this fitted function, we compute the **optimal language ratios**, and use them to train new models for the downstream evaluations reported in **Table 2**.
>
> In this workflow, the formulation of CLIMB is based entirely on randomized experiments, while its effectiveness is verified through extrapolation and downstream performance—there is **no overlap between hypothesis generation and evaluation**, and thus no circular reasoning.
>
> We appreciate the reminder that some qualitative observations in Figure 2 (e.g., diminishing returns, influence of language similarity) have been discussed in prior work. These works indeed support our empirical findings, and we will revise the paper to cite them accordingly.
>
> ---
> ### Q3.2: Why not compare with open multilingual models like XGLM?
>
> Thank you for the suggestion. We have added **XGLM-1.7B** and **XGLM-7.5B** results to Table R3.1 and Table R3.2.
>
> **Direct comparisons with open models like XGLM are not entirely fair**, due to differences in:
>
> - **Training data**: Our models use only **FineWeb-2**, while open models always include curated data such as **QA, code, math and synthetic corpora**.
> - **Training scale**: Open models are trained on **more tokens** (e.g., 26T for QWEN3), compared to **1T** for our models.
> - **Language coverage**: Open models are trained on **more languages** than CLIMB.
> - **Initialization**: Some open models benefit from **distillation from larger proprietary models**.
>
> Despite these disadvantages, CLIMB-trained models achieve **comparable or stronger results** on several multilingual benchmarks.
>
> **Table R3.1** Comparison of CLIMB with open-source baselines on 7B
> |7B|Include|MGSM|Belebele|MultiBLiMP|XNLI|XCOPA|XSC|Flores|GMMLU|CMMLU|JMMLU|VLMU|XWG|XHS|XARC-E|XARC-C|XTQA|
> |--|-------|----|--------|-----------|----|-----|----|------|-----|-----|-----|-----|----|----|-------|-------|-----|
> |Qwen3-8B-Base|**59.45**|**60.51**|**87.99**|71.18|**46.85**|**67.02**|**66.97**|56.32|**42.70**|**56.38**|**47.15**|**47.59**|**86.49**|**59.98**|**74.13**|**54.42**|**43.90**|
> |XGLM-7.5B|25.58|0.69|24.27|71.76|41.59|61.40|60.97|27.15|27.78|32.26|29.84|31.98|75.99|43.86|48.23|28.79|32.13|
> |Temp|29.64|9.93|42.19|71.14|44.86|61.73|64.44|55.15|33.51|37.33|36.78|36.68|82.72|57.12|66.32|42.70|43.97|
> |UniMax|30.76|9.67|40.90|71.10|44.85|62.16|64.88|55.57|33.97|37.25|37.19|36.47|82.40|57.88|67.75|44.28|42.41|
> |CLIMB|31.76|12.29|41.72|**72.67**|43.65|63.44|65.07|**56.47**|34.98|37.83|40.54|35.40|84.02|59.90|69.90|46.78|42.86|
>
>
> **Table R3.2** Comparison of CLIMB with open-source baselines on 1B
> |1B|Include|MGSM|Belebele|MultiBLiMP|XNLI|XCOPA|XSC|Flores|GMMLU|CMMLU|JMMLU|VLMU|XWG|XHS|XARC-E|XARC-C|XTQA|
> |--|-------|----|--------|-----------|----|-----|----|------|-----|-----|-----|-----|----|----|-------|-------|-----|
> |Llama-3.2-1B|28.48|3.89|30.69|77.66|41.29|57.07|58.13|44.14|28.74|30.30|29.84|29.09|76.08|41.52|47.19|29.55|34.91|
> |Qwen3-1.7B|**46.87**|**36.95**|**74.81**|77.72|**43.25**|59.58|60.47|50.25|**34.24**|**45.16**|**36.09**|**36.26**|**79.12**|46.59|59.53|**40.18**|38.55|
> |Gemma-3-1b-pt|25.63|1.78|28.13|**80.88**|36.33|55.58|54.84|46.55|27.12|28.59|28.48|30.05|65.79|41.59|49.29|29.69|37.32|
> |XGLM-1.7B|25.96|1.89|24.21|68.10|37.35|56.73|56.99|21.80|26.08|29.04|28.37|29.41|62.60|36.18|40.17|24.60|33.31|
> |UniMax|25.12|2.07|23.30|62.24|40.88|59.35|58.99|49.12|31.27|33.75|32.26|31.16|74.94|48.22|59.39|35.78|39.67|
> |Temp|24.82|2.11|25.27|60.75|41.03|58.69|59.54|47.46|29.04|33.03|31.98|30.27|74.76|47.56|59.31|34.93|39.38|
> |CLIMB|25.92|2.40|26.17|65.54|41.65|**59.98**|**60.54**|**50.43**|31.78|33.67|33.21|31.76|77.48|**48.75**|**60.45**|36.56|**40.94**|
>
> ---
>
> ### Q3.3: Why are the bold values in Table 2 not the best overall?
>
> Thank you for the suggestion. As noted in Line 269 and Q2, **direct comparisons with open-source models are not fully fair** due to differences in data, scale, and setup.
>
> Therefore, in Table 2 we **bold only the best results among our sampling methods** to ensure fair intra-group comparison. We will clarify this bolding strategy in the table caption in the camera-ready version.
>
> ---
> ### Q3.4: Why are standard errors and significance testing missing?
>
> Thanks for the question. We report **standard errors** (from the evaluation harness) in Table R4. However, they are consistently small across benchmarks and fail to reflect meaningful differences.
>
> To better assess significance, we conducted **paired $t$-tests** between CLIMB and each baseline. As shown in Table R5, **CLIMB shows statistically significant improvements on 13–15 out of 17 benchmarks** under both 1B2 and 7B settings (*p* < 0.05).
>
> **Table R3.3** Standard errors (stderr) from Harness for 1B2 models.
>
> |1B2 stderr|Include|MGSM|Belebele|MultiBLiMP|XNLI|XCOPA|XSC|Flores|GMMLU|CMMLU|JMMLU|VLMU|XWG|XHS|XARC-E|XARC-C|XTQA|
> |----------|-------|----|--------|-----------|----|-----|----|------|-----|-----|-----|-----|----|----|-------|-------|-----|
> |Uniform|25.19±0.003|2.11±0.003|26.17±0.003|62.24±0.001|40.08±0.003|59.49±0.007|58.99±0.002|47.51±0.067|29.05±0.001|34.79±0.004|32.47±0.005|31.44±0.015|73.34±0.007|48.12±0.001|59.76±0.002|35.41±0.003|39.62±0.003|
> |CLIMB|25.92±0.003|2.40±0.003|26.17±0.003|65.54±0.001|41.65±0.003|59.98±0.007|60.54±0.002|50.43±0.066|31.78±0.001|33.67±0.004|33.21±0.005|31.76±0.015|77.48±0.006|48.75±0.001|60.45±0.002|36.56±0.003|40.94±0.003|
>
> **Table R3.4** Significance test on 7B and 1B2 models (✅ means p < 0.05)
>
> | CLIMB v.s.|include|MGSM|Belebele|MultiBLiMP|XNLI|XCOPA|XSC|Flores|GMMLU|CMMLU|JMMLU|VLMU|XWG|XHS|XARC-E|XARC-C|XTQA|#significant (p✓)|
> |-----------|-------|----|--------|-----------|-----|------|----|-------|------|------|------|------|----|----|--------|--------|-----|-------------------|
> |**7B Models**|||||||||||||||||||||
> |Uniform|✅|0.24|✅|✅|✅|0.28|✅|✅|✅|✅|0.11|✅|✅|✅|✅|✅|✅|14/17|
> Temp|✅|✅|✅|✅|✅|✅|✅|✅|✅|✅|0.34|✅|0.92|✅|✅|✅|✅|15/17|
> |UniMax|✅|✅|0.11|✅|✅|✅|✅|✅|✅|✅|0.39|✅|✅|✅|✅|✅|✅|15/17|
> |**1B2 Models**|||||||||||||||||||||
> |Uniform|0.43|0.94|✅|✅|✅|✅|✅|✅|✅|✅|✅|✅|✅|✅|✅|✅|✅|15/17|
> |Temp|✅|0.41|✅|✅|✅|✅|✅|✅|✅|✅|0.33|0.20|✅|✅|✅|✅|✅|14/17|
> |UniMax|0.44|0.92|✅|✅|✅|✅|✅|✅|✅|✅|0.11|✅|✅|✅|0.18|✅|✅|13/17|
>
> ---
>
> ### Q3.5: Are 7B results fairly compared to other models?
>
> Thank you for highlighting this. 7B results are in Table R3 and the significance results are in Table R5.
>
> ---
>
> ### Q3.6: Is “state-of-the-art” appropriate?
>
>
> Thank you for pointing this out. We agree that the term “state-of-the-art” may be misleading in this context. We will revise this wording in the camera-ready version.
>
> ---
>
> ### Q3.7: Missing Details about Translated Benchmarks
>
> Thank you for the detailed feedback. We clarify that our translated benchmark was first generated using **GPT-4o**, and then evaluated via **human review**. Translations were rated on a 5-point scale, with **GPT outputs scoring 4.3 on average**, and **final human-reviewed versions scoring even higher**.
>
> Human evaluators were selected based on **CEFR standard**: C1 level in the target language and B2 in English. In total, human evaluation cost **$31,212**, ensuring quality and reliability.
>
> We also applied the same evaluation process to datasets from [5–9] and found our translations to be **comparable or stronger**. As a result, we prioritized **our own translated benchmark** to better align with the 16-language setup in our study.
>
> To address concerns about translation artifacts, we further added **non-translated benchmarks** (Belebele, MGSM, MultiBLiMP) in Table R3/R4. Since our model is not trained on code/math data, we excluded MCLM (competition-level math) to reflect our domain coverage.
>
> We will clarify these details in the revised version.
>
> ---
>
> ### Q3.8: Does the tokenizer design affect results?
>
> Thank you for the insightful question. We used a **custom tokenizer** to better study the link between **tokenizer design and cross-lingual transfer**. Public tokenizers (e.g., XGLM, mT5) often have opaque training distributions, making such analysis difficult.
>
> To ensure transparency, we trained our tokenizer on **6B tokens** from the 16 target languages, with **50% English** and the rest evenly split. This resulted in:
> - **English compression rate**: 21.3%
> - **Non-English average**: 29.4%
>
> We found that CLIMB remains **robust across different tokenizers**, though the fitted parameters do vary. This suggests that tokenizer design influences cross-lingual interactions, which we plan to explore further in future work.
>
> ---
>
> ### Q3.9: Which GlobalMMLU subset is used?
>
> Thank you for your question. We used **both the culturally specific and culturally agnostic subsets** of GlobalMMLU in our evaluation.
>
> ---
> ### Q3.10: Why does CLIMB underperform QWEN on GMMLU, GPQA, and TruthfulQA?
>
> Thank you for the question. This is addressed in Q3.2.

---

> > ### Comment · Reviewer_iYcL · 2025-08-04
> > **Response to Authors**
> >
> > >We confirm that there is no circularity in our experimental design.
> >
> > Could the authors clarify then what is is the relationship between the models described in Section 2 and the ones in Section 3? Is this a completely different set of models?
> >
> > >Direct comparisons with open models like XGLM are not entirely fair
> >
> > In Table 2, you compare your method to existing models like Llama, Qwen, and Gemma. All of these also differ in these four respects. Nevertheless, I appreciate the addition of these evaluations. I agree that it does not affect the comparisons between CLIMB and the other models.
> >
> > >direct comparisons with open-source models are not fully fair
> >
> > The paper makes a direct comparison between the models trained and existing models: "Despite being trained on a relatively modest 1 T-token budget, our 1.2 B-parameter model achieves competitive performance against publicly available models such as LLaMA-3.2, Gemma-3, and Qwen-3." (L284-286). In combination with the bolding strategy, this comment is misleading. It suggests that the models trained in the paper match performance with these models, when in fact Qwen significantly outperforms the model on many benchmarks. The authors should update this point in the paper and change the bolding strategy to prevent misinterpretation.
> >
> > As to the other points, if the authors update the manuscript to include the additional details provided in their rebuttal, I will be satisfied.

---

> ### Author Response · Authors · 2025-08-05
> **Response to Reviewer iYcL**
>
> We sincerely thank the reviewer for the thoughtful and constructive feedback. **We will incorporate all these experiments and clarifications in the revised version of the paper.** If any concerns remain, we would be more than happy to address them. **We sincerely hope our clarifications will support a positive assessment.**
>
> ---
> ### **1. Clarifying the Relationship Between Models in Section 2 and Section 3**
>
> Thank you for the question. We would like to clarify the relationship between the models used in Section 2 and those in Section 3.
>
> In **Section 2**, our goal is to explore how **language ratios**, **training token amounts**, and **validation loss** are related. We trained around **500 different models** with diverse language allocation settings, from which we observed consistent patterns and derived a predictive function.
>
>
> In **Section 3**, we validate whether this function can reliably capture and generalize the observed trends. This validation is carried out in two steps:
>
> - In **Section 3.2**, we evaluate the fitting accuracy of the function on **the same 500 models** from Section 2 (the "Fitting Results" in Table 1). We understand the concern that this may appear as circular validation. However, this step is similar to reporting **"loss on the training set"** in standard machine learning tasks—it helps assess whether the function has enough capacity to fit the trends. Importantly, by fitting on a **large and diverse set of 500 models**, we further reduce the risk of overfitting to specific error patterns.
>
> - In **Section 3.3**, we further validate the function on an **additional set of 40 models** that are **completely separate** from those in Section 2 (the "Extrapolation Results" in Table 1). There is no overlap between these two sets. These held-out models serve as a validation set to test generalization. Results on these held-out models demonstrate that the fitted function maintains strong performance, even on models trained with **larger token budgets**, confirming its ability to **generalize beyond the model in section 2**.
>
> Overall, this validation procedure aligns with standard scaling law methodology [1-3] and does not constitute circular validation. We hope this explanation helps clarify the experimental setup and addresses concerns about potential circularity.
>
> ---
>
> ### **2. On Misleading Comparisons with Open-Source Models (LLaMA, Qwen, Gemma)**
>
> We appreciate the reviewer’s understanding that direct comparisons with open-source models are not entirely fair. Our goal is simply to provide **a reference for readers**, not to overstate our model's performance.
>
> To avoid potential misunderstanding, we will make the following revisions:
> - Clearly state that **bolded results represent the best performance among data allocation methods** in the table caption.
> - Avoid using terms like "state-of-the-art" in the main text to prevent overclaiming.
>
> ---
>
> References:
>
> [1] Hoffmann, Jordan, et al. "Training Compute-Optimal Large Language Models." *Advances in Neural Information Processing Systems 35 (NeurIPS 2022).*
>
> [2] Que, Haoran, et al. "D-CPT Law: Domain-Specific Continual Pre-training Scaling Law for Large Language Models." *Advances in Neural Information Processing Systems 37 (NeurIPS 2024).*
>
> [3] Kaplan, Jared, et al. "Scaling laws for neural language models." *arXiv preprint arXiv:2001.08361* (2020).

---

> > ### Comment · Reviewer_iYcL · 2025-08-05
> > **Response**
> >
> > >Clarifying the Relationship Between Models in Section 2 and Section 3
> >
> > Thank you for clarifying. I think using the results from the same set of models to justify the subsequent set of experiments seems somewhat circular to me, but I think if the authors clarify this in the updated manuscript, it would mostly address my concern.
> >
> > >On Misleading Comparisons with Open-Source Models
> >
> > Again, I think if this is clarified in the updated paper, this largely addresses my concerns.
> >
> > As many of my concerns have been at least mostly addressed, I raise my score to 4.

---

> > > ### Author Response · Authors · 2025-08-06
> > >
> > > We sincerely appreciate your thoughtful review. The clarifications and additions provided in the rebuttal will be integrated into the final version. If there are any remaining concerns, we would be happy to discuss them further.

---

### Official Review · Reviewer_4qz9 · 2025-07-03

**Clarity:** 3
**Significance:** 3
**Originality:** 3
**Rating:** 4
**Confidence:** 3

**Summary:**

This paper proposes a multilingual optimization framework, called CLIMB, to address the challenges to determine the optimal language ratios in the multilingual LLM pre-training. Experimental results demonstrate that the proposed approach achieves better accuracy and consistently outperforms baselines on a variety of multilingual tasks at different size of model parameters.

- The paper is well motivated and mostly easy to follow. However, each figure is too small to read. Please enlarge them in the camera-ready.
- The authors conducted extensive experiments across different languages in a variety of multilingual tasks. The experimental results with both 1.2B and 7B models support scalability of the proposed approach. Overall, the proposed approach is technically sound.
- Have you increased the language coverage up to > 100 languages to test the massively multilingual settings? Temperature sampling is a well-known technique to balance language data during multilingual training, and I was wondering if you gave simple comparison.
- Any insights on which languages (e.g., low-resource languages) mostly benefit from this approach?

**Questions:**

see the summary.

**Ethical Concerns:**

["NO or VERY MINOR ethics concerns only"]

**Final Justification:**

Thank you for sharing the new results in response to the questions. These results should enrich the current manuscript. I am keeping my score of 4 to reflect a still learning toward acceptance, though not a strong accept (<5). As pointed out by the other reviewers, the initial manuscript lacked the critical baselines such as temperature sampling. This inclusion of these new results could be considered as major changes.

**Quality:**

3

**Strengths And Weaknesses:**

strengths
- The paper is well motivated and mostly easy to follow. However, each figure is too small to read. Please enlarge them in the camera-ready.
- Extensive experimental results. It shows that the proposed approach is technically sound.

weakness
- while the authors explores multilingual experimental settings, it'd be interesting to conduct an experiment in a massively multilingual settings with nearly 100 languages. The language coverage up to 16 languages or so look relatively less complexity.
- Along with that, it'd be nice to analyze the effectiveness of the proposed approach per language or language family

---

> ### Author Rebuttal · Authors · 2025-07-31
>
> Thank you for your thoughtful review and valuable suggestions.
>
> ---
> ### Q2.1: Figures are hard to read—can they be enlarged?
>
> Thank you for pointing this out. We will enlarge all figures and improve their readability in the camera-ready version.
>
> ---
> ### Q2.2: Did you compare CLIMB with temperature sampling?
>
> Thank you for the suggestion. We have added two strong baselines—**temperature sampling** (Temp) and **UniMax** (suggested by Reviewer 1T2b)—to our updated experiments. We also included **non-translated benchmarks** (Belebele, MGSM, MultiBLiMP, Include, suggested by Reviewer iYcL) to better assess multilingual capabilities.
>
> As shown in **Table R2.1** and **R2.2**, CLIMB consistently outperforms both baselines across benchmarks and model sizes, confirming its advantage over heuristic sampling methods.
>
> **Table R2.1** 1B2 model Performance on new baselines and new benchmarks.
> | 1B2 | include | mgsm | belebele | multiblimp | XNLI  | XCOPA | XSC   | Flores | GMMLU | CMMLU | JMMLU | VLMU  | XWG   | XHS   | XARC-E | XARC-C | XTQA  |
> |--------|---------|------|----------|-------------|-------|--------|--------|--------|--------|--------|--------|--------|--------|--------|---------|---------|--------|
> | Temp   | 24.82   | 2.11 | 25.27    | 60.75       | 41.03 | 58.69  | 59.54  | 47.46  | 29.04 | 33.03 | 31.98 | 30.27 | 74.76 | 47.56 | 59.31  | 34.93  | 39.38 |
> | UniMax | 25.12   | 2.07 | 23.30    | 62.24       | 40.88 | 59.35  | 58.99  | 49.12  | 31.27 | **33.75** | 32.26 | 31.16 | 74.94 | 48.22 | 59.39  | 35.78  | 39.67 |
> | CLIMB  | **25.92**   | **2.40** | **26.17**    | **65.54**       | **41.65** | **59.98**  | **60.54**  | **50.43**  | **31.78** | 33.67 | **33.21** | **31.76** | **77.48** | **48.75** | **60.45**  | **36.56**  | **40.94** |
>
> **Table R2.2** 7B model Performance on new baselines and new benchmarks.
> | 7B | include | mgsm | belebele | multiblimp | XNLI  | XCOPA | XSC   | Flores | GMMLU | CMMLU | JMMLU | VLMU  | XWG   | XHS   | XARC-E | XARC-C | XTQA  |
> |--------|---------|------|----------|-------------|-------|--------|--------|--------|--------|--------|--------|--------|--------|--------|---------|---------|--------|
> | Temp   | 29.64   | 9.93 | **42.19**    | 71.14       | **44.86** | 61.73  | 64.44  | 55.15  | 33.51 | 37.33 | 36.78 | **36.68** | 82.72 | 57.12 | 66.32  | 42.70  | **43.97** |
> | UniMax | 30.76   | 9.67 | 40.90    | 71.10       | 44.85 | 62.16  | 64.88  | 55.57  | 33.97 | 37.25 | 37.19 | 36.47 | 82.40 | 57.88 | 67.75  | 44.28  | 42.41 |
> | CLIMB  | **31.76**   | 12.29| 41.72    | **72.67**       | 43.65 | **63.44**  | **65.07**  | **56.47**  | **34.98** | **37.83** | **40.54** | 35.40 | **84.02** | **59.90** | **69.90**  | **46.78**  | 42.86 |
>
> ---
> ### Q2.3: Why only 16 languages? Can CLIMB scale to 100+?
>
> This is a great question. To handle massive multilinguality, we applied CLIMB at the **language family level** instead of individual languages.
>
> We filtered out families with fewer than **100B tokens** in FineWeb-2 and retained **58 language families**, covering **300+ languages** in total. Results from **Table R2.3** on the 1.2B model show that CLIMB still outperforms temperature sampling under this broader setting. What's more, CLIMB outperform open-source models on many benchmarks. Given the disadvantage of this comparison (details are listed in Rebuttal to iYcL Q3.2), this further claims the advantage over heuristic sampling methods.
>
> **Table R2.3** 1B model Performance on CLIMB-300+.
>
> | Model         | Include | MGSM  | Belebele | MultiBLiMP | XNLI  | XCOPA | XSC   | Flores | GMMLU | CMMLU | JMMLU | VLMU  | XWG   |
> |---------------|---------|-------|----------|-------------|-------|--------|--------|--------|--------|--------|--------|--------|--------|
> | Llama-3.2-1B   | 28.48   | 3.89  | 30.69    | 77.66       | 41.29 | 57.07  | 58.13  | 44.14  | 28.74 | 30.30 | 29.84 | 29.09 | 76.08 |
> | Qwen3-1.7B (distilled from model trained with 26T token)     | **46.87**   | **36.95** | **74.81**    | 77.72       | **43.25** | 59.58  | 60.47  | 50.25  | **34.24** | **45.16** | **36.09** | **36.26** | **79.12** |
> | gemma-3-1b-pt  | 25.63   | 1.78  | 28.13    | **80.88**       | 36.33 | 55.58  | 54.84  | 46.55  | 27.12 | 28.59 | 28.48 | 30.05 | 65.79 |
> | xglm-1.7B      | 25.96   | 1.89  | 24.21    | 68.10       | 37.35 | 56.73  | 56.99  | 21.81  | 26.08 | 29.04 | 28.37 | 29.41 | 62.60 |
> | Temp-300+      | 24.82   | 2.11  | 25.27    | 68.28       | 42.03 | 59.29  | 59.54  | 49.46  | 29.04 | 28.90 | 29.54 | 30.30 | 74.76 |
> | CLIMB-300+     | 25.35   | 2.22  | 27.32    | 73.21       | 42.91 | **60.04**  | 60.12  | **50.47**  | 32.57 | 33.03 | 31.98 | 32.27 | 76.78 |
> | CLIMB-16       | 25.92   | 2.40  | 26.17    | 65.54       | 41.65 | 59.98  | **60.54**  | 50.43  | 31.78 | 33.67 | 33.21 | 31.76 | 77.48 |
>
>
> We plan to further extend CLIMB to scale efficiently in even larger language scenarios.
>
> ---
>
> ### Q2.4: Can you provide per-language or per-family analysis?
>
> Thank you for the suggestion. To better understand CLIMB’s behavior across languages, we analyzed **cross-lingual interaction ratios** among the 16 languages in our setting. This helps explain how CLIMB assigns language proportions in a data-driven manner.
>
> **Table R2.3** average transferability for each language and the top-1 transfer-out language.
>
> | Language          | de    | en    | nl    | es    | pt    | fr    | it    | id    | ja    | ko    | zh    | ru    | ar    | th    | vi    | tr    |
> |--------------------|-------|-------|-------|-------|-------|-------|-------|-------|-------|-------|-------|-------|-------|-------|-------|-------|
> | Transfer-out Score | 0.123 | 0.199 | 0.130 | 0.218 | 0.113 | 0.174 | 0.146 | 0.181 | 0.124 | 0.146 | 0.121 | 0.139 | 0.144 | 0.108 | 0.066 | 0.144 |
> | Transfer-in Score  | 0.168 | 0.151 | 0.144 | 0.188 | 0.146 | 0.145 | 0.192 | 0.139 | 0.164 | 0.129 | 0.108 | 0.185 | 0.130 | 0.152 | 0.127 | 0.136 |
> | Top1_Lang     | nl     | it     | de     | pt     | es     | es     | pt     | de     | zh     | ja     | ja     | tr     | it     | vi     | th     | ru     |
>
>
> **Key observations:**
> 1. **Intra-family transfer is strong.** Languages within the same family (e.g., Spanish–Portuguese, Japanese–Korean) tend to benefit from one another, which supports CLIMB’s use of family-level grouping in large-scale allocations.
> 2. **High-transfer languages** like English, Spanish, and Indonesian receive larger allocations, as their benefits generalize well to other languages (see also Figure 7).
>
> This analysis provides interpretability for CLIMB’s language decisions and further validates its allocation strategy.
>
>
> ---
>
> ### Q2.5: Which languages benefit the most from CLIMB?
>
> Thank you for the question. To understand CLIMB’s allocation behavior, we analyze **transfer-out** and **transfer-in** scores for each language (see Table R2.3).
>
> - A **higher transfer-out score** means the language contributes more to improving other languages.
> - A **lower transfer-in score** means the language gains less from others and must rely more on its own data.
>
> We observe that **English, Spanish, and Indonesian** have the highest transfer-out scores. Adding these languages benefits many others, so CLIMB tends to allocate more to them to maximize overall utility.
>
> In contrast, **Chinese (zh)** has the **lowest transfer-in score**, indicating it gains little from cross-lingual transfer. As a result, CLIMB assigns it a higher share of training data to preserve its performance—this may also explain its relatively smaller improvements on downstream tasks.
>
> These findings are consistent with CLIMB’s final allocation in **Figure 7**, confirming that the learned language proportions are grounded in measurable cross-lingual dynamics rather than heuristics.

---

> > ### Comment · Reviewer_4qz9 · 2025-08-05
> >
> > Thank you for sharing the new results in response to the questions. These results should enrich the current manuscript. I am keeping my score of 4 to reflect a still learning toward acceptance, though not a strong accept (<5). As pointed out by the other reviewers, the initial manuscript lacked the critical baselines such as temperature sampling. This inclusion of these new results could be considered as major changes.

---

> > > ### Author Response · Authors · 2025-08-06
> > >
> > > Thank you for your constructive feedback. We will include the additional content from the rebuttal in the camera-ready version. Please feel free to reach out if you have any further questions or suggestions.

---

### Official Review · Reviewer_1T2b · 2025-07-03

**Clarity:** 3
**Significance:** 2
**Originality:** 2
**Rating:** 3
**Confidence:** 4

**Summary:**

This paper proposes CLIMB, a new method to optimally allocate different languages in the training data for large multilingual language models. It introduces a way to measure how languages interact and influence each other during training, then uses this to find the best mix of languages to improve overall model performance. Experiments show that models trained with CLIMB’s language proportions perform better or as well the baselines in the paper.

**Questions:**

The paper is clear. I have mentioned the weaknesses in the weaknesses section. The authors can address these weaknesses by responding to them as questions.

**Ethical Concerns:**

["NO or VERY MINOR ethics concerns only"]

**Final Justification:**

The authors were unfamiliar with some fundamental methods, and I essentially guided them to at least include basic comparisons.
I increased my score from 2 to 3 given their effort in the rebuttal.

**Limitations:**

yes

**Paper Formatting Concerns:**

I did not notice any major formatting issues in this paper.

**Quality:**

2

**Strengths And Weaknesses:**

Strengths:

- The approach builds on scaling laws, an influential concept in LLM research, and innovatively extends it with cross-lingual interaction metrics.
- Most of the benchmarks are multiple-choice questions; however, they include translations as well, which is a plus since they do not limit themselves to non-generative tasks.

Weaknesses:

- Weak baselines [Major reason "weak evaluation" for score 2]: The paper did not compare with more effective sampling methods such as UniMax (https://arxiv.org/pdf/2304.09151) or even simple temperature-based sampling (see the same paper).
- Most benchmarks are automated machine-translated. We definitely have benchmarks that are not, such as Belebele, covering more than 100 languages (https://arxiv.org/abs/2308.16884).
- The number of tokens is treated as a variable in the paper, but the model size (number of parameters) should also be considered as a variable (they have two sizes: 1.2B and 7B). Otherwise, conclusions might be confounded by this factor and may not hold true across different model sizes.
- The order of operation is quite high (I did not calculate the exact value; this is based on the number of nested loops). Would you mind correcting me if I am wrong and calculating the number of times training occurs for a given number of languages n?

Suggestions:

- Update the FineWeb2 citation for the camera-ready version, as it now has an arXiv paper (https://arxiv.org/abs/2506.20920).
- Missing highly related work: This paper https://arxiv.org/abs/2404.07982 shows the role of language imbalance (which you are optimizing) in cross-lingual generalization (which you are measuring).

---

> ### Author Rebuttal · Authors · 2025-07-31
>
> Thank you for your thoughtful review and valuable suggestions. **We are eager to further address any additional concerns you may have. We sincerely hope this response helps clarify the key points and merits of our work, and would greatly appreciate the opportunity to improve the evaluation of our work.**
>
> ---
>
> ### Q1.1: Weak Baselines
>
> Thank you for raising this point. We have conducted additional experiments to include two stronger baselines: **UniMax** and **temperature-based sampling (Temp)**. The results below show that our proposed method, **CLIMB**, consistently outperforms both baselines across multiple benchmarks and model sizes (1.2B and 7B):
>
> **Table R1.1** 1B2 model Performance on new baselines and new benchmarks.
> | 1B2 | include | mgsm | belebele | multiblimp | XNLI  | XCOPA | XSC   | Flores | GMMLU | CMMLU | JMMLU | VLMU  | XWG   | XHS   | XARC-E | XARC-C | XTQA  |
> |--------|---------|------|----------|-------------|-------|--------|--------|--------|--------|--------|--------|--------|--------|--------|---------|---------|--------|
> | Temp   | 24.82   | 2.11 | 25.27    | 60.75       | 41.03 | 58.69  | 59.54  | 47.46  | 29.04 | 33.03 | 31.98 | 30.27 | 74.76 | 47.56 | 59.31  | 34.93  | 39.38 |
> | UniMax | 25.12   | 2.07 | 23.30    | 62.24       | 40.88 | 59.35  | 58.99  | 49.12  | 31.27 | **33.75** | 32.26 | 31.16 | 74.94 | 48.22 | 59.39  | 35.78  | 39.67 |
> | CLIMB  | **25.92**   | **2.40** | **26.17**    | **65.54**       | **41.65** | **59.98**  | **60.54**  | **50.43**  | **31.78** | 33.67 | **33.21** | **31.76** | **77.48** | **48.75** | **60.45**  | **36.56**  | **40.94** |
>
> **Table R1.2** 7B model Performance on new baselines and new benchmarks.
> | 7B | include | mgsm | belebele | multiblimp | XNLI  | XCOPA | XSC   | Flores | GMMLU | CMMLU | JMMLU | VLMU  | XWG   | XHS   | XARC-E | XARC-C | XTQA  |
> |--------|---------|------|----------|-------------|-------|--------|--------|--------|--------|--------|--------|--------|--------|--------|---------|---------|--------|
> | Temp   | 29.64   | 9.93 | **42.19**    | 71.14       | **44.86** | 61.73  | 64.44  | 55.15  | 33.51 | 37.33 | 36.78 | **36.68** | 82.72 | 57.12 | 66.32  | 42.70  | **43.97** |
> | UniMax | 30.76   | 9.67 | 40.90    | 71.10       | 44.85 | 62.16  | 64.88  | 55.57  | 33.97 | 37.25 | 37.19 | 36.47 | 82.40 | 57.88 | 67.75  | 44.28  | 42.41 |
> | CLIMB  | **31.76**   | 12.29| 41.72    | **72.67**       | 43.65 | **63.44**  | **65.07**  | **56.47**  | **34.98** | **37.83** | **40.54** | 35.40 | **84.02** | **59.90** | **69.90**  | **46.78**  | 42.86 |
>
> In addition, we conducted statistical significance testing. Results show that the improvements of CLIMB over UniMax and temperature sampling are statistically significant (*p* < 0.05) on the majority of benchmarks. Detailed significance tests are provided in our response to Reviewer iYcL. These results confirm that CLIMB is a strong and reliable method for multilingual data allocation.
>
> ---
>
> ### Q1.2: Benchmark Quality: Human vs. Machine Translated
>
> We appreciate this feedback. To address the concern, we have included results on additional human-translated benchmarks such as **Belebele**, **Include**, **MultiBLiMP** and **MGSM** (suggested by Reviewer iYcL), as shown in Table R1.1 and R1.2. The consistent advantage of **CLIMB** on both human-translated and machine-translated benchmarks demonstrates its robustness and generalizability. We will include these results in the camera-ready version.
>
> ---
>
> ### Q1.3: Controlling for Model Size in Scaling Law
>
> Thank you for this thoughtful suggestion. Our current work primarily focuses on understanding how **training data scale** influences optimal language allocation. This is the central variable in our formulation and analysis.
>
> We also evaluated CLIMB on models of different sizes, including **530M**, **1.2B**, and **7B**, and applied the same scaling law fitting procedure. We found that the fitted parameters remain **highly consistent** across model sizes, with only minor differences—primarily in the scaling constant **$E$**, which does **not affect the final computed ratios**.
>
> We further quantified this consistency by computing the **relative standard deviation (RSD)** across all parameters, which was only **1.62%**, indicating strong robustness to model scale. In addition, as shown in **Tables R1.1/R1.2** and **Figure 6**, CLIMB performs well on both **1.2B and 7B models**, demonstrating its practical effectiveness across scales.
>
> We agree that explicitly modeling **parameter count** as an additional input to the scaling law could further improve allocation quality, and we have included this as a future direction in **Appendix G**.
>
>
> ---
>
> ### Q1.4: Complexity of CLIMB Training Procedure
>
> Thank you for raising this concern. We clarify that the overall training complexity of CLIMB is **linear in the number of languages $N$**, not exponential.
>
> As reported in Line 171, CLIMB performs the following steps:
>
> - **Monolingual Scaling Fitting**: For each language, we train **2 monolingual models** with different data scales → total: **$2 × N$**
> - **Cross-lingual Interaction Measurement**: For each language, we sample **2 random language ratios**, each evaluated under **2 different data scales** → total: **$4 × N$**
>
> **Total training runs $= 2 × N + 4 × N = 6 × N$**
>
> This **$O(N)$** complexity is manageable in practice, and we believe it is justified by the substantial improvements in convergence and downstream performance.
>
> ---
>
> ### Q1.5: Related Work on Language Imbalance and Generalization
>
> Thank you for pointing out this relevant work on language imbalance and its effect on cross-lingual generalization. We acknowledge its relevance and will include a discussion and proper citation in the camera-ready version.

---

> ### Author Response · Authors · 2025-08-04
>
> Dear Reviewer 1T2b,
>
> Thank you once again for your thoughtful and constructive feedback on our work.
>
> We hope our responses have addressed your concerns clearly. If you have any further questions or suggestions, please feel free to  let us know— we would be more than happy to provide additional clarification.
>
> We sincerely appreciate your review, which has played an important role in improving the overall quality of our paper.
>
> Best regards,
> All authors

---

> ### Comment · Reviewer_1T2b · 2025-08-05
>
> Thanks for the new results. I’ll increase my rating by 1 (2 to 3).
> I had mentioned some sampling methods (like temperature-based sampling) as potential baselines worth comparing, and I appreciate that you addressed them in your response. I also noticed that other reviewers raised similar concerns, and you put considerable effort into the rebuttal, especially in response to reviewer iYcL.
> That said, the paper would benefit from a more comprehensive review of existing methods, or at least a mention of additional ones in the related work section. This could have been a much stronger paper if the authors had demonstrated a deeper familiarity with the relevant literature.

---

> > ### Author Response · Authors · 2025-08-05
> > **Response to Reviewer 1T2b**
> >
> > Thank you very much for increasing your rating.
> >
> > To ensure we fully addressed your concerns, could you confirm **whether any issues remain regarding our experimental analyses**, or if **the remaining point is the completeness of related work citations**? If it’s the latter, we have thoroughly updated the related work and section 2.2.1 for the camera-ready version. If there are any other suggestions, please let us know. We remain committed to improving our manuscript. **We sincerely hope you will consider our paper for acceptance.**
> >
> > Additionally, we would like to briefly clarify the goals and logic of our experiments. The primary objective of our work is to **quantitatively** analyze and model cross-lingual transfer effects. To achieve this goal rigorously, our experimental design systematically controls all relevant variables, ensuring only one parameter is varied at a time. Consequently, the trends observed in Figure 2 are directly derived from our own empirical evaluations.
> >
> > We recognize and appreciate that previous studies have **qualitatively** identified similar patterns, and these earlier findings indeed serve as valuable supporting evidence for our quantitative results. **Our work takes a step further by moving from qualitative observation to quantitative modeling, providing a more systematic and measurable understanding of cross-lingual transfer.** We have carefully revised both the Empirical Observations (Section 2.2.1) and the related work section to explicitly acknowledge these prior studies, and these changes will be fully incorporated into the camera-ready manuscript.
> >
> > We greatly appreciate your insightful feedback, which has substantially enhanced the quality of our paper.
> >
> > Best regards,
> > All authors

---

> > ### Author Response · Authors · 2025-08-06
> >
> > Dear Reviewer 1T2b,
> >
> > Thank you for raising your score. As 3 is still a negative recommendation, we hope to continue the discussion and work toward a more positive assessment.
> >
> > Regarding your latest comment:
> >
> > On the experiments—you raised similar concerns as Reviewer iYcL, who has since updated their score to positive after our rebuttal. If you still have doubts, we’re happy to clarify further.
> >
> > On the related work—thank you for highlighting those papers. We’ve also reviewed additional literature and will include a more thorough comparison in the camera-ready version.
> >
> > If there are any other concerns, please feel free to let us know—we're glad to address them.
> >
> > Best regards,
> > All authors

---

> > ### Author Response · Authors · 2025-08-08
> >
> > Dear Reviewer 1T2b,
> >
> > We appreciate your earlier feedback. As you remain the only reviewer with a negative recommendation, we hope you might reconsider your rating or let us know if there are still any issues to address.
> >
> > Best regards,
> > All authors

---

> > ### Author Response · Authors · 2025-08-09
> >
> > Dear Reviewer 1T2b,
> >
> > We truly value your time and input, and we hope our responses have addressed your concerns. Any additional feedback would be greatly appreciated.
> >
> > Best regards, All authors

---

### Note · Authors · 2025-08-12

We sincerely thank all reviewers, the ACs, and the PCs for their time, constructive feedback, and engagement throughout the review process.

Our work focuses on **quantitatively modeling cross-lingual transfer in LLMs, which to our knowledge has not been systematically studied before**. To ensure accurate fitting, we invested **substantial computational resources (256 H100 GPUs over two months)**. All reviewers recognized the novelty and potential impact of this study in the multilingual modeling domain.

During the rebuttal phase, two main concerns were raised:

**Missing baseline** – Several reviewers suggested including temperature sampling as a baseline. We have added this in our revised results, and all reviewers confirmed that the concern has been addressed.

**Related work** – Two reviewers noted missing citations. We have reviewed the relevant literature in depth, identified the related studies, and will incorporate proper citations in the camera-ready version.

Beyond the two critical issues above, all other points will also be addressed in the camera-ready version. **Given that all reviewers noted the paper is clearly written and easy to follow, the overall changes will be limited in scope**, with no need for major revisions. We hope our clarifications and updates can assist in the final decision-making process.

We again thank all reviewers and chairs for their thoughtful feedback, which has helped us further strengthen the paper.

---

### Decision · Program_Chairs · 2025-09-17

**Decision:**

Accept (poster)

**Comment:**

This paper introduces CLIMB, a cross-lingual interaction-aware framework for optimizing multilingual data allocation in LLM pretraining. Extensive experiments across multiple scales show that CLIMB outperforms heuristic baselines. The work makes a strong claim of moving beyond heuristic sampling toward a principled, quantitatively grounded method for multilingual data allocation. However, weaknesses include the initial omission of strong baselines (later added during rebuttal), limited exploration of massively multilingual settings in the main paper, and presentation issues such as small figures and potentially misleading claims (around "SOTA" models).

During rebuttal, the authors addressed most major concerns, and also committed to revising misleading claims and citations. Most reviewers raised their scores after rebuttal, recognizing that the additional results significantly strengthened the paper. Weighing the novelty and technical rigor, I support acceptance of this paper, though the authors should polish the paper / fix the presentation issues as promised.